# Topology- and Gradient-Guided Knowledge Distillation for Point Cloud Semantic Segmentation

**Luu Tung Hai**                                             *luutunghai@gmail.com*
*The University of Alabama at Birmingham, United States*

**Thinh D. Le**                                             *thomlestudy295@gmail.com*
*Soongsil University, South Korea*

**Zhicheng Ding**                                             *dingz@bgsu.edu*
*Bowling Green State University, United States*

**Qing Tian**                                             *qtian@uab.edu*
*The University of Alabama at Birmingham, United States*

**Truong-Son Hy** *                                             *thy@uab.edu*
*The University of Alabama at Birmingham, United States*

**Reviewed on OpenReview:** *https://openreview.net/forum?id=lP2phGa5af*

## Abstract

Point cloud processing has gained significant attention due to its critical role in applications such as autonomous driving and 3D object recognition. However, deploying high-performance models like Point Transformer V3 in resource-constrained environments remains challenging due to their high computational and memory demands. This work introduces a novel distillation framework that leverages topology-aware representations and gradient-guided knowledge distillation to effectively transfer knowledge from a high-capacity teacher to a lightweight student model. Our approach captures the underlying geometric structures of point clouds while selectively guiding the student model's learning process through gradient-based feature alignment. Experimental results in the Nuscenes, SemanticKITTI, and Waymo datasets demonstrate that the proposed method achieves competitive performance, with an approximately $16\times$ reduction in model size and up to $1.9\times$ decrease in inference time compared to its teacher model. Notably, on NuScenes, our method achieves competitive performance among knowledge distillation techniques trained solely on LiDAR data, surpassing prior knowledge distillation baselines in segmentation performance. Our implementation is available publicly at `https://github.com/HySonLab/PointDistill`.

## 1 Introduction

Point cloud data are a critical representation of 3D geometry and have become essential in a wide range of applications, from autonomous driving and robotic navigation to urban mapping Zhou & Tuzel (2018); Geiger et al. (2012); Gomez-Ojeda et al. (2016); Oh & Watanabe (2002). Recent advances in deep learning have enabled significant progress in point cloud processing, with models such as Point Transformer V3 Wu et al. (2024) setting new benchmarks in accuracy and robustness. Despite the success of models like Point Transformer V3, their high computational demands and memory requirements Golla & Klein (2015); Cao et al. (2019) pose challenges for deployment in resource-constrained environments, such as edge devices or real-time systems. To address this issue, various model compression strategies have been introduced,

---
*Corresponding author

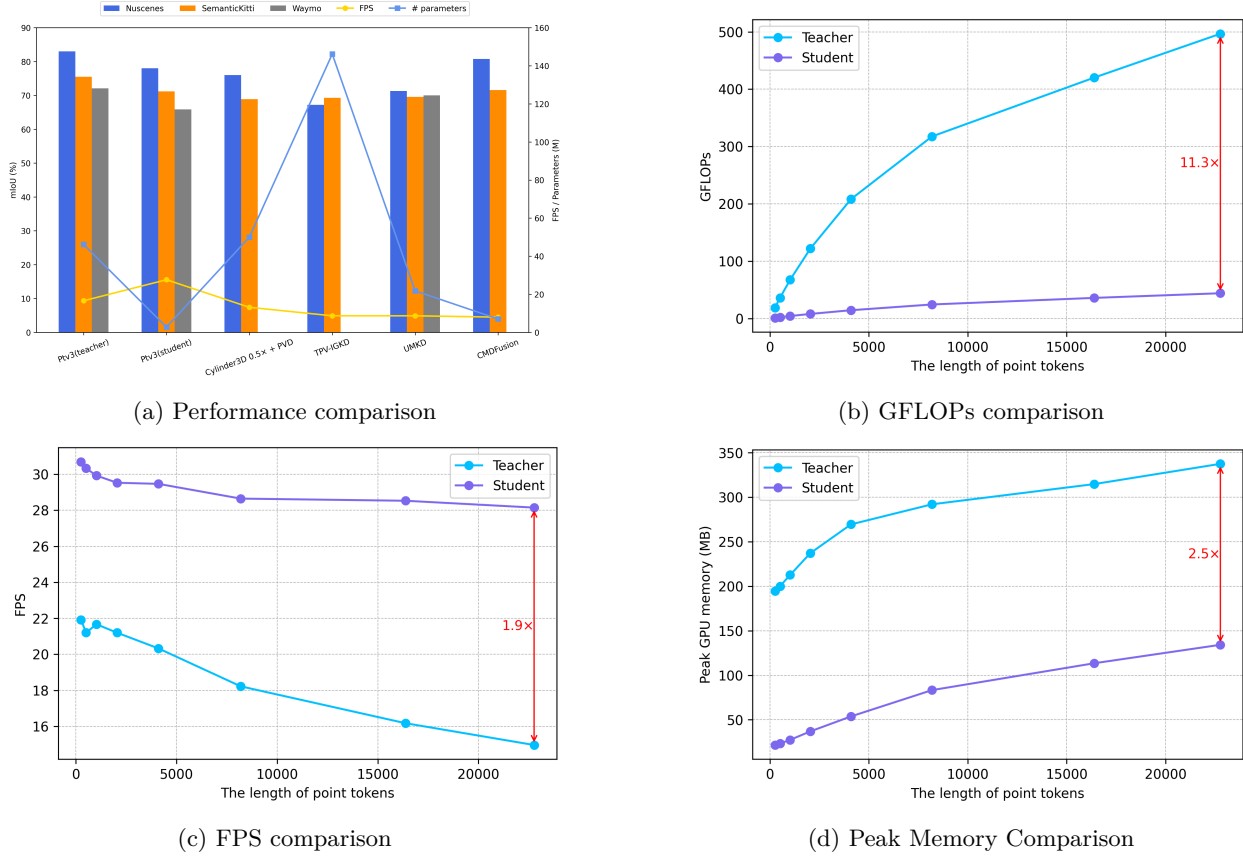

(a) Performance comparison

(b) GFLOPs comparison

(c) FPS comparison

(d) Peak Memory Comparison

Figure 1: Comprehensive comparisons between our proposed method and state-of-the-art knowledge distillation baselines across multiple evaluation metrics Hou et al. (2022c); Li et al. (2024b); Sun et al. (2024); Cen et al. (2024). (a) The radar chart demonstrates that our method achieves consistently better mIoU on three key datasets (NuScenes, SemanticKITTI, and Waymo), along with favorable FPS and memory efficiency. (b)-(d) As the input point token length increases, our approach maintains lower GPU memory usage and FLOPs, while sustaining significantly faster inference speed. (d) Peak GPU memory usage during inference, measured using max memory allocated function in Torch. This metric reflects the highest amount of memory used by the PyTorch tensors by the caching allocator during the inference phase. Notably, this value may differ significantly from the memory reported by the PyTorch Profiler, as it includes temporary allocations used by CUDA kernels.

including methods such as network pruning Han et al. (2016); Liu et al. (2019b); Louizos et al. (2018), quantization Choi et al. (2019); Dong et al. (2022); Nagel et al. (2019), lightweight model architectures Howard et al. (2019); Ma et al. (2018), and knowledge distillation Zhang et al. (2023); Hou et al. (2022a); Zhang et al. (2024).

Knowledge distillation is a machine learning technique that aims to transfer knowledge from a large and high capacity model to a smaller and more efficient model Hinton et al. (2014); Romero et al. (2015); Sanh et al. (2019). This approach allows the student model to approximate the performance of the teacher while being computationally less demanding, making it suitable for deployment in resource-constrained environments such as edge devices or mobile platforms. Over the years, knowledge distillation has been effectively applied in various domains, including image recognition Romero et al. (2015); Liu et al. (2019a) and natural language processing Hahn & Choi (2019); He et al. (2021); Rashid et al. (2021), demonstrating its versatility and impact. Recently, several approaches have been introduced to incorporate knowledge distillation into 3D detection tasks using point cloud data Guo et al. (2021b); Zhang et al. (2024). However, these methods focus primarily on multimodal student-teacher selection in a multimodal context, often overlooking the unique geometric characteristics of point clouds.

To overcome these deployment hurdles, we propose a framework combining topology-aware representations with gradient-guided distillation. The framework leverages the inherent geometric and structural properties of point clouds to preserve critical topological information during the distillation process. By integrating gradient-based guidance, the proposed approach selectively emphasizes salient geometric features that contribute most significantly to the model's performance, enabling efficient knowledge transfer from a high-capacity teacher model to a lightweight student model. This strategy ensures that the student model retains competitive accuracy while significantly reducing computational and memory requirements, making it suitable for real-time and edge-based applications.

Extensive experiments on the proposed method have been conducted to demonstrate the effectiveness of our approach over previous knowledge distillation methods. Our main contributions can be summarized as follows.

- We propose a novel distillation framework that integrates topology-aware knowledge representation and gradient-guided distillation techniques, addressing the challenges of deploying high-performance point cloud models in resource-constrained environments.

- The framework leverages the unique geometric and structural properties of point clouds, embedding topological information into the distillation process to ensure the preservation of critical features necessary for accurate predictions.

- By incorporating gradient-guided distillation, our method selectively emphasizes salient features, enabling efficient and effective knowledge transfer from the teacher model to the student model.

- Extensive experimental results on popular benchmark datasets, such as Nuscenes reveal that our approach achieves up to a $16\times$ reduction in the number of parameters and a $77.75\%$ reduction in CUDA memory consumption in linear operations and a $2.5\times$ lower in peak CUDA memory usage during inference while maintaining accuracy within $5\%$ of state-of-the-art of non-distilled methods.

While traditional knowledge distillation approaches, such as matching logits or intermediate attention features Wu et al. (2023); Han et al. (2024); Wang et al. (2020) have proven effective for compressing standard transformer architectures, they are often insufficient for 3D point cloud semantic segmentation. Under aggressive compression, lightweight 3D student networks exhibit severe geometric degradation, as (i) the fragmentation of large continuous regions (loss of connectedness) and (ii) the over-smoothing of boundaries for thin or small structures. Standard KD objectives do not explicitly preserve the global geometric continuity and critical structural regions necessary to prevent these issues. Therefore, our framework is designed to be complementary to traditional KD: rather than replacing standard distillation, we explicitly transfer structural topology (inspired by topology-preserving dense prediction methods Hu et al. (2019)) and gradient-guided spatial saliency to resolve these 3D-specific failure modes.

## 2 Related Works

**3D Point Cloud Processing**. Representing 3D data via point clouds is central to autonomous driving and robotics. Traditional deep learning approaches fall into three main categories Halperin & Eisl (2025): projection-based methods Chen et al. (2017); Lang et al. (2019); Li et al. (2016) that map points to 2D grids (often sacrificing geometric detail); voxel-based methods Choy et al. (2019); Song et al. (2017); Wang et al. (2017) using 3D sparse convolutions (limited by grid resolution and sparsity); and point-based methods Ma et al. (2022); Qi et al. (2017); Thomas et al. (2019); Zhao et al. (2019) that directly process raw points. While early point-based approaches struggled with local structures, modern transformer architectures Guo et al. (2021a); Robert et al. (2023); Wu et al. (2022b); Yang et al. (2023) have significantly improved long-range dependency modeling.

Furthermore, specialized feature extraction frameworks like MASS Li et al. (2022b) and PillarSegNet Manivasagam et al. (2021) have been developed to optimize dense top-view (BEV) and sparse pillar representations. While these designs excel at their specific representations, our work takes an orthogonal approach focused on knowledge distillation (KD). Specifically, our topology-aware KD operates in the latent feature

space, transferring structural geometry from a high-capacity point-based teacher to a lightweight student. Distilling across fundamentally different representations (e.g., point-to-voxel) requires complex spatial alignment modules. By adopting a homogeneous point-to-point setup, we isolate our topological loss from cross-representation confounding variables, leaving cross-modal distillation for future work.

**Point Transformer Architecture**. Transformers improve point-based methods by leveraging self-attention to capture multi-scale dependencies Guo et al. (2021a); Wu et al. (2022b). Evolving from Point Transformer V1 Zhao et al. (2021) and V2 Wu et al. (2022a), which suffered from high computational costs and kNN bottlenecks—Point Transformer V3 (PTv3) Wu et al. (2024) shifted toward simplicity by serializing point clouds via space-filling curves and serialized patch attention. Despite achieving state-of-the-art results with massive reductions in memory and latency, PTv3's preprocessing overhead and reliance on high-end hardware still limit its deployment in resource-constrained, real-time scenarios.

**Knowledge distillation (KD)** improves a lightweight student model by transferring knowledge from a pre-trained teacher, offering a pathway to compress heavyweight models like PTv3. Moving beyond early softmax matching Hinton et al. (2015), modern KD aligns intermediate feature layers Liu et al. (2019a); Romero et al. (2015); Tung & Mori (2019) to capture the rich geometric contexts essential for 3D point-cloud data.

**Multi-Step Knowledge Distillation**. When compressing a high-capacity teacher into a significantly smaller student, the large capacity gap can lead to optimization difficulties or training collapse. In such extreme scenarios, introducing an intermediate Teacher Assistant (TA) model (e.g., TAKD Mirzadeh et al. (2020), AMD Han et al. (2024)) is a standard multi-step solution to smooth knowledge transfer. However, our proposed framework bypasses the need for a TA model for two reasons. First, our student is a scaled-down variant of the same PTv3 backbone family, achieving a strong baseline even without distillation; thus, optimization stability is not the primary bottleneck in this regime. Second, rather than mitigating the capacity gap with intermediate networks, we explicitly target the qualitative information lost during compression. By transferring global topological structures and gradient-guided spatial saliency, our method provides the structural signals necessary to reduce reliance on an intermediate model. Nevertheless, combining our topology-aware distillation with a TA framework represents a promising direction for future work in even more aggressive compression settings.

**Topological Distillation** leverages topological data analysis (TDA) to transfer global structural features. Methods like TGD Jeon et al. (2024) and TopKD Kim et al. (2024) distill knowledge via persistence images (PI). However, they face scalability challenges due to TDA's computational cost and approximation errors during PD-to-PI conversion, limiting their effectiveness on diverse, noisy point clouds. Unlike these existing topological KD methods designed primarily for 2D images, our approach targets large-scale LiDAR segmentation by employing exact diagram-level matching and custom gradient routing through the Vietoris-Rips complex, avoiding the scaling bottlenecks of vectorization. Furthermore, our framework complements traditional feature distillation (e.g., AD-KD Wu et al. (2023), AMD Han et al. (2024), MiniLM Wang et al. (2020)) by explicitly resolving 3D-specific failure modes, such as structural fragmentation and over-smoothed boundaries that standard KD objectives fail to preserve under strong compression.

**Differentiable Topological Layers**. Integrating persistent homology end-to-end requires overcoming the non-differentiability of the discrete persistence map. Early approaches decoupled topology from optimization via post-hoc descriptors. More recent methods like Topological Autoencoders Moor et al. (2020), PersLay Carrière et al. (2020), and Differentiable Topology Layers Brüel-Gabrielsson et al. (2020) treat persistence diagrams as differentiable layers. However, they typically rely on soft approximations or kernel-based smoothing. In contrast, our approach leverages the *inverse map theorem* to route gradients directly through the exact critical edges of the Vietoris-Rips complex Poulenard et al. (2018), enabling precise structural alignment without approximation artifacts.

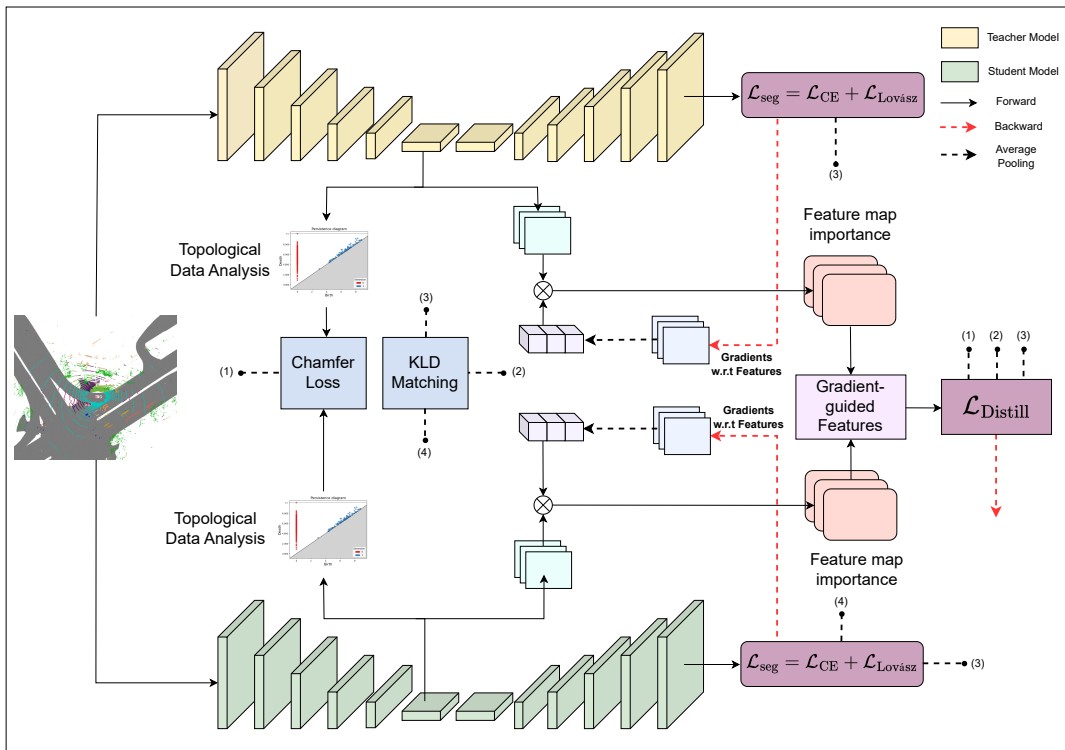

Figure 2: Overview of the proposed framework. The distillation process is driven by (1) **Topological Data Analysis (TDA)**, which aligns the persistence diagrams of feature manifolds using a weighted Chamfer distance; (2) **Logits Distillation** via KLD; and (3) **Gradient-Guided Feature Alignment**. The combination enables the lightweight student to capture global geometry ($H_1$ loops, $H_0$ clusters) and local semantics simultaneously.

## 3 Methodology

### 3.1 Overview of the Framework

Our framework introduces a topology-guided distillation approach designed to transfer both structural complexity and semantic precision from a high-capacity teacher to a lightweight student. As illustrated in Figure 2, the student mimics the teacher through a multi-objective optimization strategy. Beyond standard output replication, we introduce a Topological Distillation mechanism that aligns the underlying geometric manifolds of the feature spaces.

The student model optimizes the following objectives:

- **Targeted Topological Distillation**: We employ a differentiable Vietoris-Rips filtration to extract exact persistence diagrams ($H_0$ and $H_1$) from both models. A persistence-weighted Chamfer loss forces the student to replicate significant topological features (clusters and loops) while ignoring noise.

- **Gradient-Guided Feature Alignment**: We utilize task-specific gradients to identify and align spatially salient feature regions, ensuring the student focuses on semantically critical points.

- **Distribution Matching**: We employ Kullback-Leibler Divergence (KLD) on the soft logits to transfer the teacher's class-wise confidence distributions.

## 3.2 Manifold Alignment via Differentiable Topology

Standard losses like MSE focus on point-to-point accuracy but ignore the global shape of the data. This often causes lightweight models to learn fractured manifolds—getting the points roughly correct, but breaking the underlying geometric structure.

### 3.2.1 Differentiable Vietoris-Rips Filtration

The core challenge in integrating Topological Data Analysis (TDA) into deep learning is the discrete nature of the persistence map. Standard algorithms for computing persistent homology rely on combinatorial operations (matrix reduction over finite fields) which are non-differentiable. Integrating Topological Data Analysis (TDA) into deep learning involves computing persistent homology on the latent embeddings. Although the latent space $\mathbb{R}^C$ is continuous, for each input the network produces a finite set of feature vectors (a point cloud in $\mathbb{R}^C$). Persistent homology is explicitly designed to infer topological structure from such finite samples in a metric space, a principle widely utilized in topology-aware deep learning Moor et al. (2020); Carrière et al. (2020); Brüel-Gabrielsson et al. (2020). Furthermore, because persistent homology depends exclusively on the pairwise distance matrix, it is mathematically well-defined regardless of the high dimensionality. To ensure the topological signals remain informative and are not overwhelmed by high-dimensional distance noise, our framework relies on three design choices: (1) we apply persistent homology to learned embeddings whose geometry is inherently shaped by the segmentation task; (2) we utilize Topology-Preserving Sampling with a stability argument (Appendix A.2) Cohen-Steiner et al. (2007); and (3) we suppress near-diagonal artifacts using persistence weighting (Section 3.2.2), which reduces the impact of low-persistence noise.

The core technical challenge in this integration lies in optimization and differentiation. The standard computation of persistent homology involves discrete algorithmic steps, such as sorting simplices and matrix reduction over finite fields, which are inherently non-differentiable. To overcome this, we exploit the fact that for Vietoris–Rips filtrations the birth/death times are piecewise-smooth functions of the pairwise distances: each birth/death value equals the filtration value of a paired creator/destroyer simplex, whose filtration value is the maximum edge length within that simplex.

**Problem.** Let $f_\theta : \mathcal{X} \to \mathcal{H}$ be the encoder parameterized by $\theta$. For an input $X$, we obtain feature embeddings $H = f_\theta(X) = \{h_1, \ldots, h_N\} \subset \mathbb{R}^C$. The persistence map $\Psi : H \to \mathcal{D}$ involves sorting edge lengths and reducing a boundary matrix over a finite field $\mathbb{Z}_2$, operations which usually have zero gradients almost everywhere.

**Solution.** Although persistent homology is computed by discrete operations (sorting simplices and reducing a boundary matrix over $\mathbb{Z}_2$), the resulting diagram coordinates are piecewise-smooth functions of the underlying edge lengths: for a fixed pairing and a fixed set of maximizers defining each simplex filtration value, the birth/death times vary smoothly with the distances. Non-differentiabilities occur only at degenerate configurations, when (i) the pairing changes, or (ii) the filtration value of a critical simplex has multiple maximizing edges. Following Solomon et al. (2022); Gameiro et al. (2016); Poulenard et al. (2018), we therefore treat the Ripser Bauer (2021) as a forward-only black box and implement a custom autograd rule, which is in the forward pass we compute persistence pairs and record the corresponding birth/death critical simplices, and in the backward pass we route gradients from each pair $(b_k, d_k)$ to the critical edge(s) that realize these filtration values in the distance matrix. At degenerate configurations we use an explicit subgradient selection rule, which yields a valid gradient for stochastic optimization. In addition to the non-differentiabilities of the persistence map (pairing changes and max-edge ties), our loss uses nearest-neighbor Chamfer minima and a diagonal fallback via nested $\min(\cdot)$ operators. These are piecewise-smooth and admit valid Clarke subgradients; in implementation we backpropagate through the selected nearest neighbor (and through the diagonal term when it is active), and at ties we use a deterministic uniform averaging. This yields a valid subgradient method in stochastic optimization. Solution is described in Algorithm 1.

**Theorem 1** (Gradient Backpropagation via Critical-Edge Routing). *Let $\mathcal{H} = \{h_1, \ldots, h_N\} \subset \mathbb{R}^C$ be the set of feature embeddings for $N$ points and let $\mathbf{D} \in \mathbb{R}^{N \times N}$ is pairwise distance matrix where each entry is the Euclidean distance $\mathbf{D}_{ij} = \|h_i - h_j\|_2$. We construct a Vietoris-Rips filtration on these points. In this process, any simplex $\sigma$ (a geometric shape like a triangle or tetrahedron) appears at a specific "time" called*

*its filtration value $v(\sigma)$. This value is determined by the longest edge in the simplex:*

$$v(\sigma) = \max_{\{i,j\} \subset \sigma} \mathbf{D}_{ij}.$$

*Let a persistent homology (PH) backend (Risper Bauer (2021)) processes this filtration and returns the multiset of topological features $\mathcal{P} = \{(b_k, d_k, \dim(k))\}_{k=1}^K$. Each feature $k$ is described by: **(i)** A persistence pair $(b_k, d_k)$ representing its Birth and Death times; **(ii)** a creator simplex $\sigma_b(k)$ is the specific shape that created the feature; **(iii)** a destroyer simplex $\sigma_d(k)$ is the specific shape that destroyed (filled in) the feature.*

*To make this differentiable, we identify exactly which edges determined the birth and death times. We define the Maximizer Edge Sets as:*

$$E_b(k) := \arg \max_{\{i,j\} \subset \sigma_b(k)} \mathbf{D}_{ij}, \qquad E_d(k) := \arg \max_{\{u,v\} \subset \sigma_d(k)} \mathbf{D}_{uv}.$$

*(For $H_0$ births, $b_k = 0$ and we take $E_b(k) = \emptyset$.)*

When the filtration value of a simplex is defined by a max over its edge lengths, the gradient routes to the (edge) argmax. If the argmax is unique, the routing is unique. If several edges tie for the maximum, the function is non-smooth at that point and any Clarke subgradient is valid. Concretely, let $E_b(k)$ and $E_d(k)$ be the sets of maximizer edges for the birth and death simplices of pair $k$. We choose coefficients $\{\alpha_{ij}^{(k)}\}_{\{i,j\} \in E_b(k)}$ and $\{\beta_{uv}^{(k)}\}_{\{u,v\} \in E_d(k)}$ with $\alpha_{ij}^{(k)} \geq 0$ and $\sum_{\{i,j\} \in E_b(k)} \alpha_{ij}^{(k)} = 1$ (and similarly for $\beta$), and distribute the gradient across the tied edges accordingly. Common choice is uniform averaging over tied maximizers ($\alpha_{ij}^{(k)} = 1/|E_b(k)|$).

Let $\mathcal{L}$ be any loss differentiable with respect to the diagram coordinates $\{(b_k, d_k)\}$. Assume we are away from pairing-change degeneracies (i.e., the creator/destroyer simplices $\sigma_b(k), \sigma_d(k)$ are locally constant); at tie points among maximizer edges, interpret derivatives in the Clarke subgradient sense. Then a valid gradient of $\mathcal{L}$ with respect to $\mathbf{D}_{ij}$ (for $i < j$) is

$$\frac{\partial \mathcal{L}}{\partial \mathbf{D}_{ij}} = \sum_{k=1}^K \left( \frac{\partial \mathcal{L}}{\partial b_k} \alpha_{ij}^{(k)} \mathbb{1}_{\{\{i,j\} \in E_b(k)\}} + \frac{\partial \mathcal{L}}{\partial d_k} \beta_{ij}^{(k)} \mathbb{1}_{\{\{i,j\} \in E_d(k)\}} \right), \qquad i < j, \tag{1}$$

Where:

- $\frac{\partial \mathcal{L}}{\partial b_k}$ and $\frac{\partial \mathcal{L}}{\partial d_k}$ are the gradients from the Persistence-Weighted Chamfer Loss (PWCD) (Algorithm 3).

- $\mathbf{1}_{\{i,j\} \in E_b(k)}$ is an indicator function. It is 1 if edge $\{i,j\}$ is the Critical Creator Edge for feature $k$, and 0 otherwise.

- Symmetrically $\frac{\partial \mathcal{L}}{\partial \mathbf{D}_{ji}} = \frac{\partial \mathcal{L}}{\partial \mathbf{D}_{ij}}$.

Moreover, let $D_{ij} = \|h_i - h_j\|_2$ and let $g_{ij} := \partial L/\partial D_{ij}$ be accumulated for undirected edges $\{i,j\}$ with $i < j$ (as in Equation 1). For any stabilizer $\varepsilon_{\mathrm{dist}} > 0$, a valid gradient with respect to embeddings is obtained by summing over undirected edges: for each $i < j$,

$$\frac{\partial L}{\partial h_i} += g_{ij} \frac{h_i - h_j}{D_{ij} + \varepsilon_{\mathrm{dist}}}, \qquad \frac{\partial L}{\partial h_j} -= g_{ij} \frac{h_i - h_j}{D_{ij} + \varepsilon_{\mathrm{dist}}}.$$

In practice, Algorithm 4 computes $(\partial L/\partial b_k, \partial L/\partial d_k)$ for PWCD, and Algorithm 1 implements Equation 1 by routing these diagram gradients to the critical edges of the paired simplices, thereby obtaining $\partial L/\partial D$ and finally $\partial L/\partial F$.

### 3.2.2 Persistence-Weighted Chamfer Distance

While the Differentiable Rips Filtration successfully extracts persistence diagrams, comparing them during training presents a unique optimization challenge. Standard topological metrics, such as the Wasserstein distance, rely on optimal transport (OT). Although exact OT has cubic computational complexity, entropic regularization (e.g., Sinkhorn) Cuturi (2013) and other advanced approximations Wang et al. (2023); Zhou et al. (2022); Izquierdo & Civera (2024) make transport-based metrics practically efficient.

However, our choice of a Chamfer-style objective over Sinkhorn OT is driven primarily by gradient behavior and robustness to topological noise. In distillation, the teacher and student persistence diagrams (PD) possess dynamic, unequal cardinalities. While Unbalanced or Partial OT (UOT/POT) Chizat et al. (2018); Fatras et al. (2021) can mathematically handle unequal sets, they introduce hyperparameters (e.g., entropic regularization $\epsilon$ and mass-divergence penalties) that are notoriously difficult to tune when PD noise fluctuates across training batches. Crucially, Sinkhorn OT produces a dense, global soft matching. When the student PD contains heavy near-diagonal noise early in training, this soft matching distributes gradients broadly across many low-value matches, pulling features in sub-optimal directions.

In contrast, the Chamfer distance provides a computationally efficient $O(n^2)$-time-complexity alternative that resolves these gradient stability issues. A Chamfer formulation with a diagonal fallback yields localized, direct gradients. It naturally tolerates unequal set sizes and allows unmatched or noisy student points to be safely pushed toward the diagonal (vanishing persistence) without forcing global mass conservation (as empirically validated in Appendix D.5).

Despite these advantages, the standard Chamfer distance has a fatal flaw in knowledge distillation: it treats all target features equally. A teacher's PD often contains hundreds of low-persistence features (points near the diagonal) caused by minor fluctuations or sampling noise in the latent space. In a standard Chamfer calculation, the student would be forced to allocate representational capacity to replicate this stochastic noise, leading to overfitting of the teacher's artifacts rather than its semantic geometry.

To solve this, we introduce a *Persistence-Weighted Chamfer Distance*. This metric effectively functions as a soft attention mechanism. It forces the student to prioritize robust topological features (such as prominent loops and clusters) while safely permitting it to ignore the teacher's stochastic noise.

**Definition**. Let $P^T = \{p_i\}_{i=1}^M$ and $P^S = \{q_j\}_{j=1}^N$ be the multisets of finite persistence pairs in a fixed homology dimension, where $p = (b, d) \in \mathbb{R}^2$. Define persistence $\text{pers}(p) = d - b$, teacher weights $w(p) = \log(1 + \text{pers}(p))$, and the stabilized normalizer $\bar{w}_T = \sum_{p \in D^T} w(p) + \epsilon$. Let the squared distance to the diagonal be $\text{dist}_\Delta^2(b, d) = (d - b)^2 / 2$.

The Persistence-Weighted Chamfer Distance $\mathcal{L}_{CD}^w$ is defined as:

$$\mathcal{L}_{\text{topo}}(P^T, P^S) = \frac{1}{\bar{w}_T} \sum_{p \in P^T} w(p) \min\left( \min_{q \in P_S} \|p - q\|_2^2, \ \text{dist}_\Delta^2(p) \right) \tag{2}$$

$$+ \frac{1}{|P^S|} \sum_{q \in P^S} \min\left( \min_{p \in D_T} \|q - p\|_2^2, \ \text{dist}_\Delta^2(q) \right), \tag{3}$$

with the standard conventions that if $D^S = \emptyset$ then $\min_{q \in D_S}(\cdot) = +\infty$ and the precision term is 0, and if $D^T = \emptyset$ then the recall term is 0. This yields a well-defined objective for all $(D^T, D^S)$ and ensures that unmatched student features are penalized by moving them toward the diagonal (vanishing persistence).

**Theoretical Justification** (See Appendix A.3).

### 3.3 Saliency Aware Feature Alignment

Standard feature distillation treats all spatial locations equally, often causing the student to waste representational capacity on task irrelevant background regions rather than semantic boundaries. To address this, we introduce a Saliency Aware Feature Alignment mechanism. We interpret the gradient of the task

loss $\mathcal{L}_{\text{seg}}$ with respect to feature maps as a proxy for semantic importance, prioritizing regions where model predictions are most sensitive to feature perturbations.

### 3.3.1 Discriminative Channel Weighting

Let $F^l \in \mathbb{R}^{N \times C}$ denote the feature activations at layer $l$. We define the task sensitivity of the $k$-th channel as the global magnitude of its gradient with respect to the segmentation loss.

Although the teacher model's parameters are frozen during distillation, we explicitly compute the gradients of the task loss $\mathcal{L}_{\text{seg}}$ with respect to the *feature activations* (hidden states) to quantify spatial sensitivity. The channel-wise importance weight $\omega_l^k$ is computed as:

$$\omega_k^l = \frac{1}{N} \sum_{i=1}^{N} \left| \frac{\partial \mathcal{L}_{\text{seg}}(Y, Y_{GT})}{\partial F_{i,k}^l} \right| \tag{4}$$

where $Y_{GT}$ denotes the ground truth labels. This step identifies which feature channels are causally responsible for the correct classification, independent of the model's weight updates.

### 3.3.2 Spatial Attention Transfer

We construct a spatial saliency map $M^l \in \mathbb{R}^N$ by aggregating the weighted feature responses across the channel dimension. This aggregation collapses the feature descriptors into a scalar attention score for each point, rendering the representation dimension agnostic:

$$M_i^l = \sum_{k=1}^{C} \left| \omega_k^l F_{i,k}^l \right|. \tag{5}$$

Since the teacher and student possess different channel dimensions ($C_T \neq C_S$), the raw magnitudes of their attention maps are not directly comparable. To ensure scale invariance, we apply min max normalization to generate the final attention descriptor $\hat{M}^l$:

$$\hat{M}^l = \frac{M^l - \min(M^l)}{\max(M^l) - \min(M^l) + \epsilon}. \tag{6}$$

The alignment objective minimizes the $L_1$ distance between the normalized attention maps of the teacher ($\hat{M}_T$) and student ($\hat{M}_S$):

$$\mathcal{L}_{\text{grad}} = \frac{1}{N} \sum_{l=1}^{L} \left\| \hat{M}_T^l - \hat{M}_S^l \right\|_1. \tag{7}$$

By optimizing $\mathcal{L}_{\text{grad}}$, we force the student to replicate the teacher's *spatial focus*, ensuring that limited capacity is allocated to the most discriminative points in the cloud regardless of the underlying feature dimension.

## 3.4 Optimization Objective

To complement the geometric and spatial alignment provided by the topological and saliency modules, we incorporate a Semantic Distillation term using Kullback-Leibler Divergence (KLD) Kullback (1951). While $\mathcal{L}_{\text{topo}}$ and $\mathcal{L}_{\text{grad}}$ align the encoder's latent manifold, the KLD loss is applied to the temperature-scaled output logits of the classifier. This guides the student to mimic not only the teacher's hard predictions but also the underlying class probability distribution, capturing inter-class correlations.

The total training objective integrates the standard segmentation task with the three distillation constraints:

$$\mathcal{L}_{\text{total}} = \mathcal{L}_{\text{seg}} + \lambda_{\text{topo}} \mathcal{L}_{\text{topo}} + \lambda_{\text{grad}} \mathcal{L}_{\text{grad}} + \lambda_{\text{KLD}} \mathcal{L}_{\text{KLD}}, \tag{8}$$

where:

- $\mathcal{L}_{\text{seg}}$ is the standard Cross-Entropy loss for point-wise classification using ground truth labels.

- $\mathcal{L}_{\text{topo}}$ enforces the preservation of high-level topological structures (clusters and loops) in the latent space.

- $\mathcal{L}_{\text{grad}}$ ensures the student focuses on the same discriminative spatial regions as the teacher.

- $\mathcal{L}_{\text{KLD}}$ minimizes the KL divergence between the teacher's and student's softened logits, transferring semantic confidence.

The hyperparameters $\lambda_{\text{topo}}$, $\lambda_{\text{grad}}$, and $\lambda_{\text{KLD}}$ balance the contribution of each knowledge stream.

## 4 Experiments and Results

To assess our topology-aware distillation framework, we performed experiments on three prominent autonomous driving datasets: SemanticKITTI Behley et al. (2019), Waymo Open Dataset Sun et al. (2020), and NuScenes Caesar et al. (2020). These datasets offer large-scale, real-world point-cloud sequences, ideal for benchmarking point-cloud processing techniques. We provide detailed descriptions of datasets, training protocols, and evaluation procedures in the Appendix C.

### 4.1 Experimental Results

**Performance on nuScenes Dataset.** Table 1 compares our method against state-of-the-art frameworks on the nuScenes test set. While recent cross-modal methods integrating camera data (LC), such as TPV-IGKD Li et al. (2024a) (81.3% mIoU) and U2MKD Liu & et al. (2024) (81.2% mIoU), establish the current upper bound, our proposed LiDAR-only student achieves a highly competitive 78.2% mIoU. This significantly outperforms established baselines including Cylinder3D Zhu et al. (2021) (76.1%), AMVNet Liong et al. (2020) (76.1%), and 2DPASS Yan et al. (2022) (76.2%). The advantages of our approach are further highlighted by its class-wise performance. Notably, our student achieves the highest accuracy among all compared models, including multi-modal fusion ones in the car (94.3%) category and demonstrates superior performance in the terrain (78.8%) category. This performance directly validates our methodological design against the common failure modes of lightweight networks: structural fragmentation and boundary over-smoothing. For large geometric manifolds like terrain, our topological supervision ($\mathcal{L}_{\text{topo}}$) optimizes $H_0$ (connected components) to preserve the teacher's dominant connectedness patterns. As shown by the qualitative visualizations in Appendix D.2 (Figure 5), this effectively mitigates the feature over-smoothing that causes localized fragmentation, restoring scattered predictions into continuous semantic regions. Furthermore, the gradient-guided alignment ($\mathcal{L}_{\text{grad}}$) aligns task-loss gradient saliency with respect to hidden activations to highlight sensitive boundary regions.

**Performance on SemanticKITTI.** Table 2 details the semantic segmentation performance on the SemanticKITTI test set. Our distilled student model achieves a highly competitive 73.1% mIoU, effectively approaching the state-of-the-art MV3Dseg Sun et al. (2025) (73.2%) while using a significantly lighter architecture. Crucially, our method outperforms recent complex baselines such as KT-Weakly Wang et al. (2025) (71.4%) and C3D+SCPNet Xia et al. (2023) (71.5%). The efficacy of our distillation strategy is evidenced by a substantial 1.6% improvement over the baseline student (71.5%). Theoretical advantages of our Topology-Aware Distillation become evident in the class-wise breakdown. While multi-view methods like MV3Dseg excel in texture-heavy classes (e.g., vegetation), our approach dominates in geometrically complex and structural categories. Under aggressive compression, fine structures are typically washed out. Our persistence-based loss explicitly counters this: optimizing $H_1$ (cycles and loops) forces the student to preserve non-trivial structural patterns and thin, disconnected topological features that Scene Completion methods Song et al. (2017) tend to over-smooth to fill occlusions. Importantly, our persistence weighting mechanism isolates critical simplices, ensuring the distillation focuses strictly on these dominant, stable structures rather than chasing near-diagonal topological noise or stochastic artifacts.

We further evaluate our framework on the Waymo Open Dataset, as summarized in Table 4. Among multi-modal methods utilizing both LiDAR and camera (LC), MSeg3D Li et al. (2023) and UMKD (B) Sun et al.

| Methods | mIoU | barrier | bicycle | bus | car | construction | motorcycle | pedestrian | traffic-cone | trailer | truck | driveable | other | sidewalk | terrain | manmade | vegetation |
|---|---|---|---|---|---|---|---|---|---|---|---|---|---|---|---|---|---|
| RangeNet++ Milioto et al. (2019) | 65.5 | 66.0 | 21.3 | 77.2 | 80.9 | 30.2 | 66.8 | 69.6 | 52.1 | 54.2 | 72.3 | 94.1 | 66.6 | 63.5 | 70.1 | 83.1 | 79.8 |
| PolarNet Zhang et al. (2020b) | 71.0 | 74.7 | 28.2 | 85.3 | 90.9 | 35.1 | 77.5 | 71.3 | 58.8 | 57.4 | 76.1 | 96.5 | 71.1 | 74.7 | 74.0 | 87.3 | 85.7 |
| SalsaNext Cortinhal et al. (2020b) | 72.2 | 74.8 | 34.1 | 85.9 | 88.4 | 42.2 | 72.4 | 72.2 | 63.1 | 61.3 | 76.5 | 96.0 | 70.8 | 71.2 | 71.5 | 86.7 | 84.4 |
| Cylinder3D Zhu et al. (2021) | 76.1 | 76.4 | 40.3 | 91.2 | 93.8 | 51.3 | 78.0 | 78.9 | 64.9 | 62.1 | 84.4 | 96.8 | 71.6 | 76.4 | 75.4 | 90.5 | 87.4 |
| C3D_0.5× + KA Hou et al. (2022b) | 73.9 | 74.2 | 36.3 | 88.5 | 87.6 | 47.1 | 76.9 | 78.3 | 63.5 | 57.6 | 83.4 | 94.9 | 70.3 | 73.8 | 73.2 | 88.4 | 86.3 |
| AMVNet Liong et al. (2020) | 76.1 | 79.8 | 32.4 | 87.4 | 90.4 | 62.5 | 81.9 | 75.3 | 72.3 | 83.5 | 65.1 | 97.4 | 67.0 | 78.8 | 74.6 | 90.8 | 87.9 |
| 2DPASS Yan et al. (2022) | 76.2 | 75.3 | 43.5 | 95.3 | 91.2 | 54.5 | 78.9 | 78.2 | 62.1 | 70.0 | 84.2 | 96.3 | 73.2 | 74.2 | 74.9 | 89.8 | 85.9 |
| SDSeg3D Li et al. (2022a) | 77.7 | 77.5 | 49.4 | 93.9 | 92.5 | 54.9 | 86.7 | 80.1 | 67.8 | 65.7 | **86.0** | 96.4 | 74.0 | 74.9 | 74.5 | 86.0 | 82.8 |
| RPVNet Xu et al. (2021) | 77.6 | 78.2 | 43.4 | 92.7 | 93.2 | 49.0 | 85.7 | 80.6 | 66.9 | 69.4 | 80.5 | 96.9 | 73.5 | 75.9 | 76.0 | 90.6 | 88.9 |
| GFNet Qiu et al. (2022) | 76.1 | 81.1 | 31.6 | 76.0 | 90.5 | 60.2 | 80.7 | 75.3 | 71.8 | 82.5 | 65.1 | **97.8** | **80.4** | 80.4 | 76.2 | **91.8** | 88.9 |
| SVASeg Zhao et al. (2022) | 74.7 | 74.1 | 44.5 | 88.4 | 86.6 | 48.2 | 72.4 | 72.3 | 61.3 | 57.5 | 75.7 | 96.3 | 70.7 | 74.7 | 74.6 | 87.3 | 86.9 |
| U2MKD Liu & et al. (2024) (LC) | 81.2 | **85.9** | 43.4 | 93.1 | 90.1 | 72.5 | 86.3 | 81.6 | **78.5** | 85.7 | 76.5 | 97.6 | 69.8 | 80.3 | 77.0 | **91.8** | 89.5 |
| KT-Weakly Wang et al. (2025)(LC) | 80.2 | - | - | - | - | - | - | - | - | - | - | - | - | - | - | - | - |
| MV3Dseg Sun et al. (2025) | 81.1 | 79.7 | 59.2 | **96.0** | 92.6 | 60.3 | 88.4 | 83.6 | 70.8 | 80.2 | 81.8 | 96.7 | 74.4 | 79.6 | 76.1 | 90.0 | 88.2 |
| TPV-IGKD Li et al. (2024a) (LC) | 81.3 | 85.4 | 43.2 | 92.9 | 93.2 | **75.7** | 77.4 | 83.4 | 77.2 | **86.8** | 77.4 | 97.7 | 71.4 | **81.3** | 77.4 | 91.7 | 89.0 |
| **Teacher (PTv3)** | **83.0** | 83.5 | **59.9** | **96.0** | **95.0** | 58.3 | **91.0** | **86.4** | 77.7 | 79.2 | 85.1 | 97.2 | 77.0 | 80.9 | **80.0** | 91.2 | **90.1** |
| **Student PTv3 (w/o KD)** | 76.1 | 76.1 | 46.7 | 89.9 | 92.2 | 40.4 | 83.9 | 78.3 | 63.2 | 68.2 | 81.8 | 96.3 | 72.8 | 73.6 | 75.4 | 89.7 | 88.8 |
| **Student PTv3 (Ours)** | 78.2 | 79.1 | 48.3 | 92.9 | 94.3 | 41.3 | 85.7 | 82.9 | 62.4 | 70.2 | 80.3 | 96.7 | 76.3 | 74.2 | 78.8 | 90.0 | 89.3 |

Table 1: Comparison with state-of-the-art semantic segmentation methods on the nuScenes test set. Methods denoted with (LC) utilize LiDAR-camera fusion, whereas all others, including our proposed framework, are strictly LiDAR-only. In our setup, both the teacher and student models are based on the Point Transformer V3 (PTv3) architecture. The baseline student (w/o KD) is trained using standard Cross-Entropy (CE) loss, while our method (Ours) employs the full distillation objective detailed in Section 3.4. With competitive performance, our model requires fewer parameters, achieves higher inference speed (FPS), and consumes less memory (see Tables 3, 8, and 9 for detailed efficiency analyses).

(2024) demonstrate strong performance with validation mIoU scores of 69.6 and 71.1, respectively. Within the LiDAR-only (L) category, LidarMultiNet Ye et al. (2022) achieves a leading test mIoU of 71.1; however, it should be noted that this approach relies on 3D bounding boxes as an additional supervision signal during training. Our teacher model sets a high performance ceiling for LiDAR-only methods with a test mIoU of 71.3. Notably, our distilled student model achieves a test mIoU of 69.5, making it competitive with several heavier architectures and even approaching the performance of multi-modal methods. This performance is particularly significant given the student's efficiency; by explicitly targeting the global-structure and boundary-saliency problems inherent to strong compression, our model maintains competitive generalization while offering a substantial reduction in parameter count, as shown in Table 3.

## 4.2 Effectiveness of Individual Loss Components

Table 5 demonstrates the effectiveness of each proposed loss component. To isolate the impact of our contributions, our baseline model already incorporates standard traditional KD, combining the segmentation task loss with KL-divergence logit distillation. This traditional KD baseline achieves reasonable performance (76.9% on nuScenes, 71.8% on SemanticKITTI, and 67.9% on Waymo). However, because standard KD does not explicitly penalize structural fragmentation or boundary over-smoothing, its performance ceiling is limited under strong compression. The addition of topology ($\mathcal{L}_{\text{topo}}$) and gradient-guided alignment ($\mathcal{L}_{\text{grad}}$) consistently improves performance across all datasets over standard KD alone. This provides strong empirical evidence that while traditional KD is a necessary foundation, explicitly transferring geometric and boundary information is essential for optimal 3D point cloud compression.

| Methods | mIoU | car | bicycle | motorcycle | truck | other-vehicle | person | bicyclist | motorcyclist | road | parking | sidewalk | other-ground | building | fence | vegetation | trunk | terrain | pole | traffic |
|---|---|---|---|---|---|---|---|---|---|---|---|---|---|---|---|---|---|---|---|---|
| SalsaNext Cortinhal et al. (2020b) | 59.5 | 91.9 | 48.3 | 38.6 | 38.9 | 31.9 | 60.2 | 59.0 | 19.4 | 91.7 | 63.7 | 75.8 | 29.1 | 90.2 | 64.2 | 81.8 | 63.6 | 66.5 | 54.3 | 47.4 |
| KPConv Thomas et al. (2019) | 58.8 | 96.0 | 32.0 | 42.5 | 33.4 | 44.3 | 61.5 | 61.6 | 11.8 | 88.8 | 61.3 | 72.7 | 31.6 | **95.0** | 64.2 | 84.8 | 69.2 | 69.1 | 56.4 | 47.4 |
| FusionNet Zhang et al. (2020a) | 61.3 | 95.3 | 47.5 | 37.7 | 41.8 | 34.5 | 59.5 | 56.8 | 11.9 | 91.8 | 68.7 | 77.1 | 30.5 | 90.5 | 69.4 | 84.5 | 69.8 | 68.5 | 60.4 | 46.2 |
| KPRNet Kochanov et al. (2020) | 63.1 | 95.5 | 54.1 | 47.9 | 23.6 | 42.6 | 65.9 | 65.0 | 16.5 | 93.2 | **73.9** | 80.6 | 30.2 | 91.7 | 64.8 | 85.7 | 69.8 | 71.2 | 58.7 | 64.1 |
| TORNADONet Gerdzhev et al. (2021) | 63.1 | 94.2 | 51.2 | 48.1 | 40.0 | 38.2 | 63.6 | 60.1 | 34.9 | 89.7 | 66.7 | 74.5 | 28.7 | 91.3 | 65.8 | 85.6 | 71.5 | 70.1 | 58.0 | 49.2 |
| SPVNAS Tang et al. (2020) | 66.4 | 97.3 | 51.5 | 50.8 | 59.8 | 58.8 | 65.7 | 62.5 | 43.7 | 90.2 | 67.6 | 75.2 | 16.9 | 91.3 | 65.9 | 86.1 | 73.4 | 71.0 | 64.6 | 66.9 |
| Cylinder3D Zhu et al. (2021) | 68.9 | 97.1 | 67.6 | 50.8 | 50.8 | 58.5 | 73.7 | 69.2 | 48.0 | 92.2 | 65.0 | 77.0 | 32.3 | 90.7 | 66.5 | 85.6 | 72.5 | 69.8 | 62.4 | 66.2 |
| PolarNet+M2S-KD Qiu et al. (2023) | 58.0 | 93.5 | 45.9 | 36.3 | 27.6 | 34.9 | 55.0 | 51.4 | 15.8 | 91.1 | 64.7 | 73.8 | 26.1 | 92.5 | 67.0 | 84.6 | 63.4 | 67.4 | 50.7 | 59.5 |
| C3D+M2S-KD Qiu et al. (2023) | 65.6 | 96.4 | 60.8 | 54.8 | 42.8 | 51.2 | 69.1 | 67.8 | 34.8 | 92.2 | 66.5 | 76.7 | 30.4 | 91.1 | 65.7 | 85.5 | 69.8 | 68.6 | 60.7 | 61.0 |
| C3D+SCPNet Xia et al. (2023) | 71.5 | 97.5 | 60.9 | 56.3 | 58.6 | 65.9 | 70.7 | 71.8 | 58.7 | 93.6 | 72.1 | 80.9 | 36.2 | 93.3 | 72.1 | 86.2 | 74.1 | 71.6 | 66.7 | 71.8 |
| U2MKD Liu & et al. (2024) (LC) | 69.6 | - | - | - | - | - | - | - | - | - | - | - | - | - | - | - | - | - | - | - |
| TPV-IGKD Li et al. (2024a) | 69.3 | 96.7 | 63.3 | 65.8 | 54.6 | 63.1 | 76.5 | **80.7** | 41.6 | 90.0 | 60.3 | 75.4 | 29.1 | 91.1 | 66.3 | 85.9 | 74.2 | 70.3 | 63.8 | 69.1 |
| KT-Weakly Wang et al. (2025)(LC) | 71.4 | - | - | - | - | - | - | - | - | - | - | - | - | - | - | - | - | - | - | - |
| MV3Dseg Sun et al. (2025) | 73.2 | **97.6** | 69.8 | **76.1** | **65.4** | 61.3 | 79.1 | 80.0 | **71.1** | 94.2 | 67.6 | 78.6 | 37.5 | 91.8 | 65.9 | 85.7 | 72.8 | 73.7 | 63.5 | 60.0 |
| **Teacher (PTv3)** | **75.5** | 97.5 | **74.2** | 62.5 | 60.1 | **67.8** | 80.1 | 76.5 | 60.2 | **96.2** | 72.1 | **82.4** | **47.5** | 93.2 | **74.1** | **89.5** | **78.2** | **76.1** | **72.1** | **75.9** |
| **Student PTv3 (w/o KD)** | 71.5 | 96.0 | 71.0 | 56.5 | 56.5 | 62.2 | 77.2 | 72.8 | 56.8 | 95.5 | 68.7 | 80.5 | 36.4 | 94.0 | 70.1 | 89.0 | 76.0 | 73.4 | 66.1 | 69.8 |
| **Student PTv3 (Ours)** | 73.1 | 96.8 | 72.6 | 58.8 | 58.8 | 65.1 | 77.6 | 73.9 | 56.5 | 92.8 | 70.5 | 80.3 | 43.6 | 91.6 | 71.7 | 87.4 | 76.6 | 74.4 | 68.3 | 71.6 |

Table 2: Comparison with state-of-the-art semantic segmentation methods on the SemanticKITTI test set. Methods denoted with (LC) utilize LiDAR-camera fusion, whereas all others, including our proposed framework, are strictly LiDAR-only. In our setup, both the teacher and student models are based on the Point Transformer V3 (PTv3) architecture. The baseline student (w/o KD) is trained using standard Cross-Entropy (CE) loss, while our method (Ours) employs the full distillation objective detailed in Section 3.4. With competitive performance, our model requires fewer parameters, achieves higher inference speed (FPS), and consumes less memory (see Tables 3, 8, and 9 for detailed efficiency analyses).

| Method | Params (M) | FPS |
|---|---|---|
| RangeNet++ Milioto et al. (2019) | 50.0 | 12.5 |
| PolarNet Zhang et al. (2020b) | 45.0 | 16.7 |
| SalsaNext Cortinhal et al. (2020b) | 6.7 | 23.8 |
| Cylinder3D Zhu et al. (2021) | 53.0 | 12.0 |
| KPConv Thomas et al. (2019) | 15.0 | 12.0 |
| SPVNAS Tang et al. (2020) | 1.0 | 16.0 |
| KT-weakly Wang et al. (2025) | - | 25.0 |
| MV3DSeg Sun et al. (2025) | 45.6 | 22.7 |
| U2MKD Liu & et al. (2024) | 55.8 | 10.64 |
| TPV-IGKD Li et al. (2024a) | 146.2 | 8.7 |
| GFNet Qiu et al. (2022) | 87.6 | - |
| KPRNet Kochanov et al. (2020) | 213.2 | - |
| Teacher (PTv3) | 46.16 | 16.61 |
| **Student PTv3 (Ours)** | **2.78** | **27.64** |

Table 3: Comparison of model size (in M parameters) and inference speed (FPS) on NuScenes.

| Method | Input | mIoU (test / val) |
|---|---|---|
| MSeg3D Li et al. (2023) | LC | 70.5 / 69.6 |
| UMKD (B) Sun et al. (2024) | LC | 70.0 / 71.1 |
| PMF Zhuang et al. (2021) | LC | − / 58.2 |
| SalsaNext Cortinhal et al. (2020a) | L | 55.8 / − |
| Realsurf Sun et al. (2024) | L | 67.6 / − |
| SPVCNN++ Tang et al. (2020) | L | 67.7 / − |
| VueNet3D Sun et al. (2024) | L | 68.6 / − |
| SphereFormer Lai et al. (2023) | L | − / 69.9 |
| LidarMultiNet Ye et al. (2022) | L | 71.1 / 69.9 |
| **Student PTv3 (w/o KD)** | L | 67.2 / 66.5 |
| **Student PTv3 (Ours)** | L | **69.5 / 68.7** |
| **Teacher (PTv3)** | L | 71.3 / 69.8 |

Table 4: Semantic segmentation mIoU on Waymo Open Dataset. Inputs: LiDAR (L), Camera (C).

## 5 Conclusion and Future Work

We presented a novel distillation framework that bridges the gap between high-capacity 3D perception models and resource-constrained deployment environments. By integrating Topological Distillation with Saliency-Aware Feature Alignment, our method enables a lightweight student model to inherit not only the semantic precision of a teacher but also its underlying geometric manifold structure.

Extensive experiments on nuScenes, SemanticKITTI, and Waymo demonstrate that our approach achieves competitive performance among LiDAR-based distillation methods. Specifically, our student model retains competitive segmentation accuracy while delivering a 16× reduction in parameters and up to 1.9× acceler-

| $\mathcal{L}_{\mathrm{KLD}}$ | $\mathcal{L}_{\mathrm{seg}}$ | $\mathcal{L}_{\mathrm{topo}}$ | $\mathcal{L}_{\mathrm{grad}}$ | S.KITTI | Waymo | nuScenes |
|:---:|:---:|:---:|:---:|:---:|:---:|:---:|
| ✓ | ✓ | | | 71.8 | 67.9 | 76.9 |
| ✓ | ✓ | ✓ | | 72.6 | 68.7 | 77.6 |
| ✓ | ✓ | | ✓ | 72.3 | 68.2 | 77.2 |
| ✓ | ✓ | ✓ | ✓ | **73.1** | **69.5** | **78.2** |

Table 5: Influence of each loss component on the final performance (mIoU %).

ation across varying token lengths; 1.6× on the standard nuScenes setup. Crucially, our qualitative analysis reveals that enforcing topological consistency effectively mitigates the fragmentation common in lightweight networks, ensuring robust performance on complex, large-scale structures.

**Future Work.** Although our topology-preserving subsampling substantially reduces the cost of persistent-homology computation during training, a natural next step is to replace heuristic selection with learnable, task-adaptive topological token sampling and amortized PH computation, enabling larger filtrations and higher-order homology under the same budget. A second direction is extending topology- and gradient-guided distillation to spatio-temporal perception and multi-parameter topology, enforcing consistency across multi-sweep sequences and across geometric scales. Finally, we will study topology-aware distillation in cross-modal and open-world settings, distilling structural priors from camera/BEV and vision(-language) foundation models into deployment-oriented backbones (e.g., hardware-optimized attention or linear-time state-space models) to improve long-tail generalization and robustness under missing or misaligned modalities.

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

## A  Theoretical Foundations and Justifications

### A.1  Topology of Feature vs. Topology of Spatial Coordinates

A critical distinction in our framework is the choice to distill the topology of latent feature spaces rather than physical spatial coordinates. This design is grounded in two fundamental geometric learning principles:

1. **Semantic vs. Physical Proximity:** Input spatial coordinates are constrained by Euclidean distance, capturing only physical proximity. In contrast, deep latent features evolve to capture semantic proximity. As demonstrated by Wang et al. (2019) in Dynamic Graph CNNs (DGCNN), feature-space neighborhoods in deeper layers group semantically similar objects (e.g., aggregating features from both the front and rear tires of a vehicle) even when they are geometrically distant. This allows the network to bypass physical distance, creating semantically coherent clusters that coordinate-based topology cannot capture.

2. **Invariance to Rigid Transformations:** Coordinate-based distillation objectives (e.g., Mean Squared Error on point coordinates) penalize any deviation in spatial alignment, making them sensitive to rotation and translation. Conversely, the topology of the feature space when characterized by persistent homology is theoretically invariant to such isometries. The Stability Theorem of persistence diagrams Cohen-Steiner et al. (2007) guarantees that topological descriptors depend solely on pairwise distances, rendering them invariant to the rigid transformation of the input. By aligning feature topology, we force the student to learn the intrinsic shape of the teacher's semantic manifold Moor et al. (2020), ensuring robust knowledge transfer that generalizes across geometric perturbations Jeon et al. (2024).

### A.2  Subsampling and Stability of Vietoris–Rips Persistence

A major bottleneck in topological distillation is the computational cost of persistent homology (PH), which scales superlinearly with the number of points. We therefore compute PH on a subsample selected by Topology-Preserving Sampling (TPS) (Algorithm 5).

Let $H = \{h_1, \ldots, h_N\} \subset \mathbb{R}^C$ be the full feature set and let $H_I \subset H$ be the TPS subsample of size $M$. Define the (one-sided) covering radius

$$\varepsilon := \max_{x \in H} \min_{y \in H_I} \|x - y\|_2.$$

Because $H_I \subset H$, this equals the Hausdorff distance $d_H(H, H_I)$.

**Proposition 1 (Rips interleaving under Hausdorff perturbations).** Let $A, B$ be finite subsets of a metric space $(\mathcal{X}, d)$ with $d_H(A, B) \leq \varepsilon$. Then the Vietoris–Rips filtrations $\mathrm{Rips}(A, r)$ and $\mathrm{Rips}(B, r)$ are $2\varepsilon$-interleaved: for all $r \geq 0$,

$$\mathrm{Rips}(A, r) \to \mathrm{Rips}(B, r + 2\varepsilon) \quad \text{and} \quad \mathrm{Rips}(B, r) \to \mathrm{Rips}(A, r + 2\varepsilon),$$

via the nearest-neighbor correspondence. Consequently, their persistence modules are $2\varepsilon$-interleaved, and the bottleneck distance between the corresponding persistence diagrams in any fixed homology dimension is at most $2\varepsilon$.

**Proof.** For any edge $(a, a')$ in $\mathrm{Rips}(A, r)$, choose $b, b' \in B$ with $d(a, b) \leq \varepsilon$ and $d(a', b') \leq \varepsilon$. By the triangle inequality, $d(b, b') \leq d(b, a) + d(a, a') + d(a', b') \leq r + 2\varepsilon$, so $(b, b')$ is an edge in $\mathrm{Rips}(B, r + 2\varepsilon)$. This extends to all simplices, yielding a simplicial map. The reverse direction is analogous.

**Implication for TPS.** TPS directly minimizes the covering radius $\varepsilon$ for a fixed budget $M$, so Proposition A.1 implies that the persistence diagrams computed on the TPS subset are a controlled approximation of the persistence diagrams of the full feature set, with error bounded (in bottleneck distance) by $O(\varepsilon)$. This motivates using TPS to reduce PH cost while preserving the dominant topological structures.

### A.3 Noise Suppression via Persistence Weighting

This subsection provides the theoretical proof and Lemma 1 for Section 3.2.2, demonstrating how the logarithmic weighting function bounds the gradient influence of low-persistence features. It mathematically establishes that as the persistence of a topological feature approaches zero (representing noise), its contribution to the loss vanishes, thereby preventing the student model from overfitting to stochastic artifacts in the teacher's latent space.

**Lemma 1** (Noise Suppression via Persistence Weighting). *Let $\mathcal{D}^T$ and $\mathcal{D}^S$ be teacher and student persistence diagrams in a fixed homology dimension, containing only* finite *pairs. Define the persistence of a point $p = (b, d)$ as $\mathrm{pers}(p) = d - b$, and define the teacher weight*

$$w(p) = \log(1 + \mathrm{pers}(p)) \geq 0, \tag{9}$$

*together with the stabilized normalizer*

$$\bar{w}_T = \sum_{p \in \mathcal{D}^T} w(p) + \epsilon, \qquad \epsilon > 0. \tag{10}$$

*Consider the weighted teacher-to-student Chamfer term*

$$\mathcal{L}_{T \to S} = \frac{1}{\bar{w}_T} \sum_{p \in \mathcal{D}^T} w(p) \min_{q \in \mathcal{D}^S} \|p - q\|_2^2. \tag{11}$$

*Then for any teacher point $p_0 \in \mathcal{D}^T$ with $\mathrm{pers}(p_0) = \delta$, its contribution to $\mathcal{L}_{T \to S}$ is bounded by*

$$0 \leq \frac{w(p_0)}{\bar{w}_T} \min_{q \in \mathcal{D}^S} \|p_0 - q\|_2^2 \leq \frac{\log(1 + \delta)}{\epsilon} \min_{q \in \mathcal{D}^S} \|p_0 - q\|_2^2. \tag{12}$$

*In particular, as $\delta \to 0$ (near-diagonal/noisy features), $\log(1 + \delta) \to 0$ and this bound vanishes, so low-persistence teacher points have negligible influence on the weighted alignment term.*

*Proof.* The contribution of the Teacher's noise point $p_{noise}$ to the Recall loss is given by:

$$L_p = d_{\min} \cdot \frac{w(p_{noise})}{\bar{w}_T} \tag{13}$$

Substituting the weight definition $w(p) = \log(1 + \delta)$ and the normalization factor:

$$L_p = d_{\min} \cdot \frac{\log(1 + \delta)}{\sum_k w(k) + \epsilon} \tag{14}$$

Using the first-order Taylor expansion $\log(1 + x) \approx x$ for small $\delta$, and observing that the denominator is strictly bounded below by $\epsilon$ (since $w(k) \geq 0$):

$$L_p \approx d_{\min} \cdot \frac{\delta}{\bar{w}_T} \leq d_{\min} \cdot \frac{\delta}{\epsilon} \tag{15}$$

Taking the limit as the persistence $\delta \to 0$:

$$\lim_{\delta \to 0} L_p \leq \lim_{\delta \to 0} \left( \frac{d_{\min}}{\epsilon} \cdot \delta \right) = 0 \tag{16}$$

Since $\bar{w}_T$ is bounded away from zero ($\bar{w}_T \geq \epsilon$), the denominator does not vanish. Consequently, as the persistence $\delta \to 0$, the loss contribution $L_p \to 0$ linearly. This ensures that the Student model is not penalized for ignoring negligible artifacts in the Teacher's latent space, allowing it to focus its representational capacity on matching persistent geometric structures.

$\square$

# B  Implementation and Algorithms

## B.1  Differentiable Feature-Space Distillation

This subsection details the complete forward and backward pass of the topological distillation framework (Algorithm 1). It outlines the specific sequence of operations—including feature normalization, Topology-Preserving Sampling (TPS), and the calculation of pairwise distances—required to route gradients through the critical edges of the Vietoris-Rips filtration.

---

**Algorithm 1** Differentiable Feature-Space Topological Distillation (with Subgradient Routing)

---

**Require:** Student features $\mathbf{H}^S \in \mathbb{R}^{B \times N \times C_S}$, Teacher features $\mathbf{H}^T \in \mathbb{R}^{B \times N \times C_T}$
**Require:** Subsample size $M$ (1024), stabilizer $\epsilon > 0$, tie tolerance $\tau_{\text{tie}}$
**Ensure:** Topological Loss $\mathcal{L}_{topo}$
1: $\mathcal{L}_{topo} \leftarrow 0$
2: **for** $k = 1$ to $B$ **do**
3:     *// Step 1: Feature Normalization*
4:     $\tilde{\mathbf{H}}^S \leftarrow \text{Normalize}(\mathbf{H}^S[k])$
5:     $\tilde{\mathbf{H}}^T \leftarrow \text{Normalize}(\mathbf{H}^T[k])$
6:     *// Step 2: Topology-Preserving Selection (See Algorithm 5)*
7:     $I_k \leftarrow \text{TPS}(\tilde{\mathbf{H}}^T, M)$
8:     $\mathbf{S} \leftarrow \tilde{\mathbf{H}}^S[I_k], \quad \mathbf{T} \leftarrow \tilde{\mathbf{H}}^T[I_k]$
9:     *// Step 3: Distance Matrix Computation*
10:     $\mathbf{D}^S \leftarrow \text{PairwiseDist}(\mathbf{S}), \quad \mathbf{D}^T \leftarrow \text{PairwiseDist}(\mathbf{T})$
11:     *// Step 4: Topological Summarization (Forward with Tie-breaking)*
12:     $\{\mathcal{D}_d^S\}_{d \in \{0,1\}} \leftarrow \text{DiffRipser}(\mathbf{D}^S; \tau_{\text{tie}})$
13:     $\{\mathcal{D}_d^T\}_{d \in \{0,1\}} \leftarrow \text{Ripser}(\mathbf{D}^T)$
14:     *// Step 5: Persistence Weighted Chamfer Distance (See Algorithm 2)*
15:     **for** $d \in \{0, 1\}$ **do**
16:         $\mathcal{L}_{topo} \leftarrow \mathcal{L}_{topo} + \text{PWCD}(\mathcal{D}_d^T, \mathcal{D}_d^S; \epsilon)$
17:     **end for**
18: **end for**
19: **return** $\mathcal{L}_{topo}/B$

---

## B.2  Persistence-Weighted Chamfer Loss

This subsection defines the specialized loss function used to align the persistence diagrams of the student and teacher. It explains the logarithmic weighting mechanism designed to suppress low-persistence sampling noise while prioritizing robust topological features like loops and clusters during optimization (Algorithm 2).

## B.3  Implementation of Gradient Routing

While persistent homology is computed by discrete operations (sorting simplices and boundary-matrix reduction), the birth/death times are piecewise-smooth functions of the underlying pairwise distances. For the Vietoris–Rips filtration (simplex filtration value equals the maximum edge length), each birth/death value is realized by one or more maximizer edges in the distance matrix. At degenerate configurations where multiple edges tie for the same filtration value, the gradient is not unique; we use a standard subgradient convention by averaging the gradient uniformly over all tied maximizer edges. This yields stable optimization and avoids relying on fragile uniqueness assumptions. The method is described in Algorithm 3.

## B.4  Complexity Reduction via Topology-Preserving Selection

To implement efficient subsampling, we employ Topology-Preserving Sampling (TPS), which is a greedy approximation to the $k$-center problem. TPS explicitly minimizes the covering radius (and thus the Hausdorff

---

**Algorithm 2** Persistence-Weighted Chamfer Loss

---

**Require:** Teacher diagram $\mathcal{P}^T = \{p_i\}_{i=1}^M \subset \mathbb{R}^2$; Student diagram $\mathcal{P}^S = \{q_j\}_{j=1}^N \subset \mathbb{R}^2$; stabilizer $\epsilon > 0$.
**Ensure:** Topology loss $\mathcal{L}_{\text{topo}}$.
 1: *// 1) Trivial case*
 2: **if** $M = 0 \wedge N = 0$ **then**
 3:     **return** 0
 4: **end if**
 5: *// 2) Teacher weights*
 6: **if** $M > 0$ **then**
 7:     $WT[i] \leftarrow log(1 + max(p_i^{(2)} - p_i^{(1)}, 0))$                    # $p = (b, d)$
 8:     $\bar{W}_T \leftarrow \sum_{i=1}^M W_T[i] + \epsilon$
 9: **end if**
10: *// 3) Helper: squared distance to diagonal*
11: $\text{dist}_\Delta^2(b, d) \leftarrow \frac{(d-b)^2}{2}$
12: *// 4) If both non-empty, compute pairwise costs*
13: **if** $M > 0 \wedge N > 0$ **then**
14:     $C_{ij} \leftarrow \frac{1}{2}\|p_i - q_j\|_2^2$                    # $M \times N$
15: **end if**
16: *// 5) Recall term (Teacher $\to$ Student) with diagonal fallback*
17: **if** $M = 0$ **then**
18:     $L_{\text{recall}} \leftarrow 0$
19: **else if** $N = 0$ **then**
20:     $L_{\text{recall}} \leftarrow \frac{1}{\bar{W}_T} \sum_{i=1}^M W_T[i] \, \text{dist}_\Delta^2(p_i^{(1)}, p_i^{(2)})$
21: **else**
22:     $\min_{T \to S}[i] \leftarrow \min_j C_{ij}$
23:     $L_{\text{recall}} \leftarrow \frac{1}{\bar{W}_T} \sum_{i=1}^M W_T[i] \, \min\left\{\min_{T \to S}[i], \text{dist}_\Delta^2(p_i^{(1)}, p_i^{(2)})\right\}$
24: **end if**
25: *// 6) Precision term (Student $\to$ Teacher) with diagonal fallback*
26: **if** $N = 0$ **then**
27:     $L_{\text{prec}} \leftarrow 0$
28: **else if** $M = 0$ **then**
29:     $L_{\text{prec}} \leftarrow \frac{1}{N} \sum_{j=1}^N \text{dist}_\Delta^2(q_j^{(1)}, q_j^{(2)})$
30: **else**
31:     $\min_{S \to T}[j] \leftarrow \min_i C_{ij}$
32:     $L_{\text{prec}} \leftarrow \frac{1}{N} \sum_{j=1}^N \min\left\{\min_{S \to T}[j], \text{dist}_\Delta^2(q_j^{(1)}, q_j^{(2)})\right\}$
33: **end if**
34: **return** $L_{\text{recall}} + L_{\text{prec}}$

---

distance) between the selected subset and the full feature set. This directly reduces the worst-case approximation error of geometric/topological summaries under a fixed budget. We use TPS to select indices, then compute PH only on the selected subset. The algorithm is described in Algorithm 5.

**Quantitative Efficiency Analysis.** To validate the computational efficiency of our hybrid pipeline (combining the Ripser Bauer (2021), critical-edge gradient routing, and TPS), we benchmark the total forward-and-backward runtime against a naive pure-PyTorch implementation. As shown in Table 6, the naive approach suffers from combinatorial complexity explosion, becoming computationally intractable beyond $N = 100$ and exhausting GPU memory at $N = 200$. In contrast, our method maintains real-time performance, achieving speedup factors exceeding $6700\times$ at $N = 100$ and scaling efficiently to $N = 1024$ with negligible overhead.

---

**Algorithm 3** Critical-Edge Gradient Routing

---

**Require:** Student features $F \in \mathbb{R}^{N \times C}$; teacher diagrams $\{P_T^{(d)}\}_{d \in \{0,1\}}$; stabilizers $\varepsilon_{\text{dist}}, \varepsilon_w > 0$; tie tolerance $\tau_{\text{tie}}$.
**Ensure:** Feature gradients $\nabla_F \mathcal{L}$.
 1: Compute distance matrix $D$: $D_{ij} \leftarrow \|F_i - F_j\|_2$ for $1 \le i < j \le N$.
 2: Initialize edge accumulator $G \in \mathbb{R}^{N \times N}$ with zeros.
 3: **for** $d \in \{0, 1\}$ **do**
 4:     *// Forward pass: compute diagrams and extract critical simplices*
 5:     Run Ripser($D$) to obtain student diagram $P_S^{(d)}$ and pairing map $\Pi^{(d)}$.
 6:     Compute diagram gradients: $(g_b, g_d) \leftarrow \text{PWCDGrad}(D_T^{(d)}, D_S^{(d)}; \varepsilon_w)$ *// See algorithm 4.*
 7:     **for** $k = 1$ **to** $|D_S^{(d)}|$ **do**
 8:         Let $(\sigma_{\text{birth}}, \sigma_{\text{death}}) \leftarrow \Pi^{(d)}(k)$.
 9:         *// Route birth gradient (only for $H_1$)*
10:         **if** $d = 1$ **then**
11:             Let $\{i, j\} = \sigma_{\text{birth}}$.
12:             $G_{ij} \leftarrow G_{ij} + g_b[k]$.
13:         **end if**
14:         *// Route death gradient*
15:         **if** $d = 0$ **then**
16:             Let $\{u, v\} = \sigma_{\text{death}}$ be the destroying edge.
17:             $G_{uv} \leftarrow G_{uv} + g_d[k]$.
18:         **else**
19:             *// For $H_1$, death is caused by a triangle $\{a, b, c\}$ completing a cycle*
20:             Let $E_\Delta = \{(a, b), (b, c), (a, c)\}$.
21:             $m \leftarrow \max_{(u,v) \in E_\Delta} D_{uv}$.
22:             $E^\star \leftarrow \{(u, v) \in E_\Delta : |D_{uv} - m| \le \tau_{\text{tie}}\}$.
23:             **for** each $(u, v) \in E^\star$ **do**
24:                 $G_{uv} \leftarrow G_{uv} + g_d[k]/|E^\star|$.
25:             **end for**
26:         **end if**
27:     **end for**
28: **end for**
29: *// Backpropagate distances to features*
30: Initialize $\nabla_F \mathcal{L} \leftarrow \mathbf{0}$.
31: **for** $1 \le i < j \le N$ **do**
32:     **if** $G_{ij} \ne 0$ **then**
33:         $u_{ij} \leftarrow (F_i - F_j)/(D_{ij} + \varepsilon_{\text{dist}})$.
34:         $\nabla_{F_i} \mathcal{L} \leftarrow \nabla_{F_i} \mathcal{L} + G_{ij} \cdot u_{ij}$.
35:         $\nabla_{F_j} \mathcal{L} \leftarrow \nabla_{F_j} \mathcal{L} - G_{ij} \cdot u_{ij}$.
36:     **end if**
37: **end for**
38: **return** $\nabla_F \mathcal{L}$.

---

# C  Experimental Setup and Hyperparameters

## C.1  Datasets

**SemanticKITTI** Behley et al. (2019) provides LiDAR point clouds from urban and suburban scenes, featuring 22 sequences with dense semantic annotations across 19 classes (e.g., vehicles, pedestrians, roads). Its high resolution and detailed labels make it a rigorous testbed for semantic segmentation.

**NuScenes** Caesar et al. (2020) integrates LiDAR, camera, and radar data in 1,000 diverse scenes, including urban roads and highways. With 3D bounding box annotations for 23 object types, it challenges models with varied weather, occlusions, and dynamic elements suited for detection and segmentation tasks. In addition, we use nuScenes-lidar seg, which is an extension of nuScenes. This dataset has semantic labels of 32 categories and annotates each point from keyframes in nuScenes. We used the 700 scenes in the training

---

**Algorithm 4** PWCDGrad: Diagram-space gradients for Persistence-Weighted Chamfer Distance

---

1: **Function** PWCDGrad($D_T, D_S, \varepsilon_w$):
2:    **Input:** Teacher $D_T = \{p_i\}_{i=1}^M$, Student $D_S = \{q_k\}_{k=1}^K$, stabilizer $\varepsilon_w$
3:    **Output:** Gradient arrays $g_b, g_d$ of size $K$
4:    Initialize $g_b \leftarrow \mathbf{0}, \ g_d \leftarrow \mathbf{0}$
5:    Define $w(b, d) = \log(1 + \max(d - b, 0))$
6:    Compute $\bar{w}_T = \varepsilon_w + \sum_{i=1}^M w(b_i^T, d_i^T)$

7:    *// 1. Recall Term (Teacher $\rightarrow$ Student)*
8:    **if** $M > 0$ **and** $K > 0$ **then**
9:      **for** $i = 1$ **to** $M$ **do**
10:        $k^\star \leftarrow \arg\min_k \|q_k - p_i\|_2^2$    *// Find nearest student*
11:        $dist_{match} \leftarrow \|q_{k^\star} - p_i\|_2^2$
12:        $dist_{diag} \leftarrow \frac{1}{2}(d_i^T - b_i^T)^2$
13:        **if** $dist_{match} \leq dist_{diag}$ **then**
14:          $s \leftarrow w(b_i^T, d_i^T)/\bar{w}_T$
15:          $g_b[k^\star] \mathrel{+}= 2s \cdot (b_{k^\star} - b_i^T)$
16:          $g_d[k^\star] \mathrel{+}= 2s \cdot (d_{k^\star} - d_i^T)$
17:        **end if**
18:      **end for**
19:    **end if**

20:    *// 2. Precision Term (Student $\rightarrow$ Teacher)*
21:    **if** $K > 0$ **then**
22:      **for** $k = 1$ **to** $K$ **do**
23:        $q_k \leftarrow (b_k, d_k)$
24:        $is\_matched \leftarrow$ **false**
25:        **if** $M > 0$ **then**
26:          $i^\star \leftarrow \arg\min_i \|q_k - p_i\|_2^2$    *// Find nearest teacher*
27:          $dist_{match} \leftarrow \|q_k - p_{i^\star}\|_2^2$
28:          $dist_{diag} \leftarrow \frac{1}{2}(d_k - b_k)^2$
29:          **if** $dist_{match} \leq dist_{diag}$ **then**
30:            $g_b[k] \mathrel{+}= \frac{2}{K}(b_k - b_{i^\star}^T)$
31:            $g_d[k] \mathrel{+}= \frac{2}{K}(d_k - d_{i^\star}^T)$
32:            $is\_matched \leftarrow$ **true**
33:          **end if**
34:        **end if**
35:        *// Fallback to diagonal if no match found or teacher empty*
36:        **if not** $is\_matched$ **then**
37:          $\delta \leftarrow d_k - b_k$
38:          $g_b[k] \mathrel{-}= \frac{1}{K}\delta$
39:          $g_d[k] \mathrel{+}= \frac{1}{K}\delta$
40:        **end if**
41:      **end for**
42:    **end if**
43:    **return** $(g_b, g_d)$

---

set with segmentation labels to fine-tune for the semantic segmentation task, and the 150 scenes in the validation set to verify the performance.

**Waymo Open Dataset** Sun et al. (2020) delivers high-resolution LiDAR data from 1,000 segments in various locations, with frequent sweeps and 3D annotations for vehicles, pedestrians and cyclists. Its long-range scans and varied conditions test robustness and generalization.

### C.2   Model Configuration and Training

We use Point Transformer V3 Wu et al. (2024) as the backbone. The student model is approximately 40-50% of the block depth (2.33$\times$ fewer encoder blocks; 2$\times$ fewer decoder blocks), trained from scratch.

---

**Algorithm 5** Topology-Preserving Sampling (TPS)

---

**Require:** Feature set $\mathcal{H} = \{h_1, \ldots, h_N\}$, target size $M$
**Ensure:** Indices $I \subset \{1, \ldots, N\}$ with $|I| = M$
1: Initialize $I \leftarrow \{i_1\}$ (random index); $D_{\min}[i] \leftarrow \|h_i - h_{i_1}\|_2$
2: **for** $k = 2$ to $M$ **do**
3: $\quad i_k \leftarrow \arg\max_i D_{\min}[i]$
4: $\quad I \leftarrow I \cup \{i_k\}$
5: $\quad D_{\min}[i] \leftarrow \min\big(D_{\min}[i], \|h_i - h_{i_k}\|_2\big) \quad \forall i$
6: **end for**
7: **return** $I$

---

| | Naive Baseline | | | Ours | | | |
|---|---|---|---|---|---|---|---|
| **Input** ($N$) | **Fwd (s)** | **Bwd (s)** | **Mem (MB)** | **Fwd (s)** | **Bwd (s)** | **Mem (MB)** | **Speedup** |
| 30 | 19.31 | 0.12 | 17.1 | 0.24 | 0.002368 | 8.2 | $79\times$ |
| 60 | 321.12 | 0.32 | 40.2 | 0.05 | 0.000250 | 16.5 | $6,039\times$ |
| 100 | 963.30 | 1.06 | 128.6 | 0.14 | 0.000254 | 17.3 | $6,743\times$ |
| 200 | *Infeasible (Timeout)* | | | 0.36 | 0.000271 | 22.7 | — |
| 400 | *Infeasible (Timeout)* | | | 0.30 | 0.000265 | 29.1 | — |
| 800 | *Infeasible (Timeout)* | | | 0.40 | 0.000236 | 42.0 | — |
| 1024 | *Infeasible (Timeout)* | | | 0.37 | 0.000258 | 49.1 | — |

Table 6: **Computational Efficiency Benchmark.** Comparison of Forward (Fwd) and Backward (Bwd) pass runtimes and Peak Memory usage. While the Naive implementation suffers from cubic complexity, becoming infeasible beyond $N = 100$, our method maintains sub-second latency even at $N = 1024$.

- **Encoder Depths:** Teacher $(2, 2, 2, 6, 2) \rightarrow$ Student $(1, 1, 1, 2, 1)$.

- **Encoder Channels:** Teacher $(32, 64, 128, 256, 512) \rightarrow$ Student $(16, 16, 32, 64, 128)$.

- **Attention Heads:** Scaled down from Teacher $(2, 4, 8, 16, 32)$ to Student $(1, 1, 2, 4, 8)$.

**Hyperparameter Selection.** Exhaustively grid-searching the optimal distillation parameters across all three large-scale datasets from scratch is highly computationally demanding. Therefore, our selection strategy was to perform a targeted sweep using the nuScenes training split, selecting weights by validation mIoU; then we reused the same weights unchanged for SemanticKITTI and Waymo. We evaluated a grid where $\lambda_{\text{topo}} \in \{0.5, 4.0, 8.0\}$, and $\lambda_{\text{grad}}, \lambda_{\text{kd}} \in \{0.5, 1.0, 2.0\}$. The optimal configuration discovered on nuScenes ($\lambda_{\text{topo}} = 8.0$, $\lambda_{\text{grad}} = 1.0$, $\lambda_{\text{kd}} = 1.0$) was subsequently applied to SemanticKITTI and Waymo. The detailed grid search results on the nuScenes dataset are provided in Table 7.

**Training Protocol.** Models are trained for 50 epochs (batch size 12) using AdamW (LR 0.002, Weight Decay 0.005) with a OneCycleLR scheduler to ensure stable convergence and effective generalization. We employ a two-stage distillation strategy: the teacher is pre-trained to convergence before guiding the student.

### C.3 Data Augmentation and Resources

We apply random rotation ($\pm 1°$), uniform scaling ($0.9 - 1.1\times$), random flipping, and Gaussian jittering ($\sigma = 0.005$) before grid sampling ($0.05\,\text{m}$). All experiments were conducted on a University HPC Cluster node equipped with 2 NVIDIA A100 GPUs (81 GB VRAM).

### C.4 Detailed Efficiency and Memory Profiling

Tables 8 and 9 compare the Teacher (46.16M params) and Student (2.78M params) models on nuScenes, showcasing the significant efficiency benefits of our distillation framework. The Teacher model possesses

| $\lambda_{\text{topo}}$ | $\lambda_{\text{grad}}$ | $\lambda_{\text{KLD}}$ | mIoU |
|---|---|---|---|
| 0.5 | 0.5 | 0.5 | 77.0 |
| 0.5 | 0.5 | 1.0 | 77.2 |
| 0.5 | 0.5 | 2.0 | 77.1 |
| 0.5 | 1.0 | 0.5 | 77.1 |
| 0.5 | 1.0 | 1.0 | 77.3 |
| 0.5 | 1.0 | 2.0 | 77.2 |
| 0.5 | 2.0 | 0.5 | 76.9 |
| 0.5 | 2.0 | 1.0 | 77.1 |
| 0.5 | 2.0 | 2.0 | 77.0 |
| 4.0 | 0.5 | 0.5 | 77.5 |
| 4.0 | 0.5 | 1.0 | 77.7 |
| 4.0 | 0.5 | 2.0 | 77.6 |
| 4.0 | 1.0 | 0.5 | 77.6 |
| 4.0 | 1.0 | 1.0 | 77.8 |

| $\lambda_{\text{topo}}$ | $\lambda_{\text{grad}}$ | $\lambda_{\text{KLD}}$ | mIoU |
|---|---|---|---|
| 4.0 | 1.0 | 2.0 | 77.7 |
| 4.0 | 2.0 | 0.5 | 77.3 |
| 4.0 | 2.0 | 1.0 | 77.6 |
| 4.0 | 2.0 | 2.0 | 77.5 |
| 8.0 | 0.5 | 0.5 | 77.8 |
| 8.0 | 0.5 | 1.0 | 78.0 |
| 8.0 | 0.5 | 2.0 | 77.9 |
| 8.0 | 1.0 | 0.5 | 77.9 |
| **8.0** | **1.0** | **1.0** | **78.2** |
| 8.0 | 1.0 | 2.0 | 78.0 |
| 8.0 | 2.0 | 0.5 | 77.7 |
| 8.0 | 2.0 | 1.0 | 77.9 |
| 8.0 | 2.0 | 2.0 | 77.8 |

Table 7: Ablation study of distillation loss weights on the nuScenes dataset.

16.6× more parameters than the Student, necessitating a deeper architecture suited for high-accuracy tasks on high-end hardware. In contrast, the Student's lightweight design significantly reduces computational overhead, achieving a 36.70× reduction in encoder FLOPs. This efficiency translates to a 2.47× faster total CPU time (203.10 ms vs. 501.11 ms) and a 4.19× faster total CUDA time (102.07 ms vs. 427.31 ms), making it highly suitable for real-time applications.Additionally, the Student requires 4.5× less peak CUDA memory for matrix multiplication operations (3.57 GB vs. 16.05 GB), which allows it to benefit more effectively from Flash Attention optimizations. At the operational level, the Teacher's resource demands are markedly higher, requiring between 2.91× and 4.69× more memory for operations such as Alloc, Idx, and LN. While the Teacher provides an accuracy ceiling on high-performance hardware, the Student's drastically reduced memory footprint and accelerated execution make it ideal for deployment on resource-constrained edge devices.

| Metric | Teacher (46.16M) | Student (2.78M) | Comparison |
|---|---|---|---|
| Total Parameters | 46,160,000 ($\sim$46.16M) | 2,780,000 ($\sim$2.78M) | Student is 16.6× smaller |
| Encoder Depths | (2, 2, 2, 6, 2) | (1, 1, 1, 2, 1) | 2.33× fewer blocks |
| Encoder Channels | (32, 64, 128, 256, 512) | (16, 16, 32, 64, 128) | 2–4× smaller |
| Encoder Attention Heads | (2, 4, 8, 16, 32) | (1, 1, 2, 4, 8) | 2–4× fewer |
| Decoder Depths | (2, 2, 2, 2) | (1, 1, 1, 1) | 2× fewer blocks |
| Decoder Channels | (64, 64, 128, 256) | (64, 64, 128, 128) | Last stage is 2× smaller |
| Decoder Attention Heads | (4, 4, 8, 16) | (2, 2, 4, 8) | 2× fewer |
| Patch Size | 1024 | 1024 | Same |
| Encoder GFLOPs | 380.25 | 10.36 | 36.7× lower |
| Decoder GFLOPs | 116.44 | 33.45 | 3.48× lower |
| Attention Compute (Encoder) | 22.58 | 0.60 | 37.63× lower |
| Inference Time (excl. overhead) | $\sim$0.0592 s | $\sim$0.0362 s | 1.64× faster |
| Batch Inference Time | $\sim$7.34 s | $\sim$4.38 s | Student faster |
| FPS | $\sim$16.6 | $\sim$27.6 | $\approx$ 1.64× higher |
| Fixed Overhead | $\sim$0.018 s | $\sim$0.011 s | $\approx$ 1.64× faster |
| Attention Mechanism | Flash Attention | Flash Attention | Student benefits more |

Table 8: Comparison of Teacher and Student models on NuScenes.

| Metric | Teacher | Student | Improvement |
|---|---|---|---|
| Total CPU Time[b] | 501.11 ms | 203.10 ms | 2.47× faster |
| Total CUDA Time | 427.31 ms | 102.07 ms | 4.19× faster |
| $\text{MM}_{\text{add}}$ (CUDA Mem)[a] | 16.05 GB | 3.57 GB | 4.5× less |
| $\text{MM}_{\text{add}}$ (Self CUDA)[a] | 173.51 ms | 34.75 ms | 5.0× faster |
| Alloc (CUDA Memory) | 8.73 GB | 3.00 GB | 2.91× less |
| Idx (CUDA Memory) | 7.27 GB | 1.78 GB | 4.08× less |
| GELU (CUDA Memory) | 6.82 GB | 1.67 GB | 4.08× less |
| LN (CUDA Memory)[a] | 4.62 GB | 985.43 MB | 4.69× less |
| Infer (Self CPU Time) | 47.01 ms | 37.24 ms | 1.26× faster |
| $\text{MM}_{\text{add}}$ (Self CPU Time)[a] | 5.40 ms | 2.72 ms | 1.99× faster |

[a] Attention-related ops: matrix multiplications for $Q$, $K$, $V$ and layer normalization.
[b] Total CPU time is from a profiling run; separate memory-focused run shows 3.02× ratio (Teacher: 278.80 ms, Student: 92.44 ms).

Table 9: Comparison of memory and time usage between Teacher and Student.

# D  Additional Ablation and Sensitivity Analysis

## D.1  Visualization of Prediction

To intuitively assess the effectiveness of our distillation framework, we visualize the semantic segmentation predictions on the nuScenes validation set in Figure 3. The comparison includes the Ground Truth (a), the Teacher's prediction (b), and the Student's prediction (c). As observed, the Teacher model (b) generates high-fidelity predictions that closely align with the Ground Truth (a), particularly in capturing complex structural layouts such as road boundaries and vehicle clusters. Crucially, the Student model (c), despite its reduced capacity, successfully replicates this performance. The Student accurately preserves the global topology of the scene. Large-scale connected components, such as the drivable surface (brown) and the surrounding road network, are segmented with high continuity, which is effectively minimizing fragmentation. This contributes to our Topological Distillation, which enforces the preservation of persistent geometric features like loops and clusters Hu et al. (2019). Additionally, small and scattered classes, such as vegetation (green points) and thin linear structures like barriers (gray lines), are sharply resolved. Standard distillation methods often fail to capture these high-frequency details due to limited receptive fields or effective context aggregation Liu et al. (2019a). The sharpness of these boundaries in Figure 3c confirm that our Saliency-Aware Alignment successfully focused the Student's attention on these spatially discriminative regions, preventing them from being washed out by the dominant background classes.

Figure 4 illustrates how persistent homology captures the evolution of topological features across different filtration scales. Given a point cloud, we construct a simplicial complex and track the birth and death of topological structures as the filtration parameter $\epsilon$ increases. The persistence diagram $D = \{(b_i, d_i)\}_{i=1}^{M}$ quantifies these events, where each point represents a topological feature. Longer bars correspond to persistent structures that encode essential geometric patterns, while shorter bars typically represent noise or minor perturbations.

Visualizing the persistence diagrams allows us to better understand the types of geometric features captured by the teacher model, such as connected components ($H_0$), loops ($H_1$) and voids ($H_2$). By encouraging the student to mimic these persistent topological features through topology-aware distillation, we aim to transfer not only semantic knowledge but also the critical underlying geometric structures necessary for robust point-cloud understanding. This visualization supports the intuition behind our method, showing that topological summaries can effectively reflect meaningful geometric information beyond what is captured by the Euclidean feature alignment.

## D.2  Boundary Visualization and Failure Cases

**Qualitative Improvements.** We visualise cropped regions where the baseline lightweight student (with standard KD) severely fractures continuous spatial structures and exhibits boundary discontinuity. Our

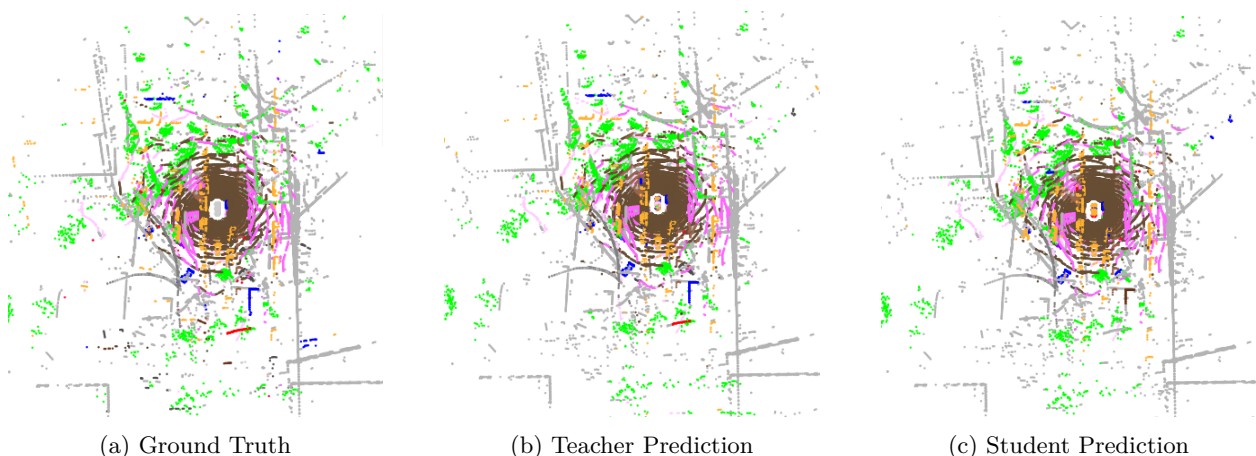

(a) Ground Truth        (b) Teacher Prediction        (c) Student Prediction

Figure 3: Visualization of our method on the nuScenes validation set. (a) Ground truth, (b) teacher model prediction, and (c) student model prediction. The student model closely follows the teacher's output and ground truth, successfully capturing almost all object classes, demonstrating the effectiveness of the knowledge distillation process.

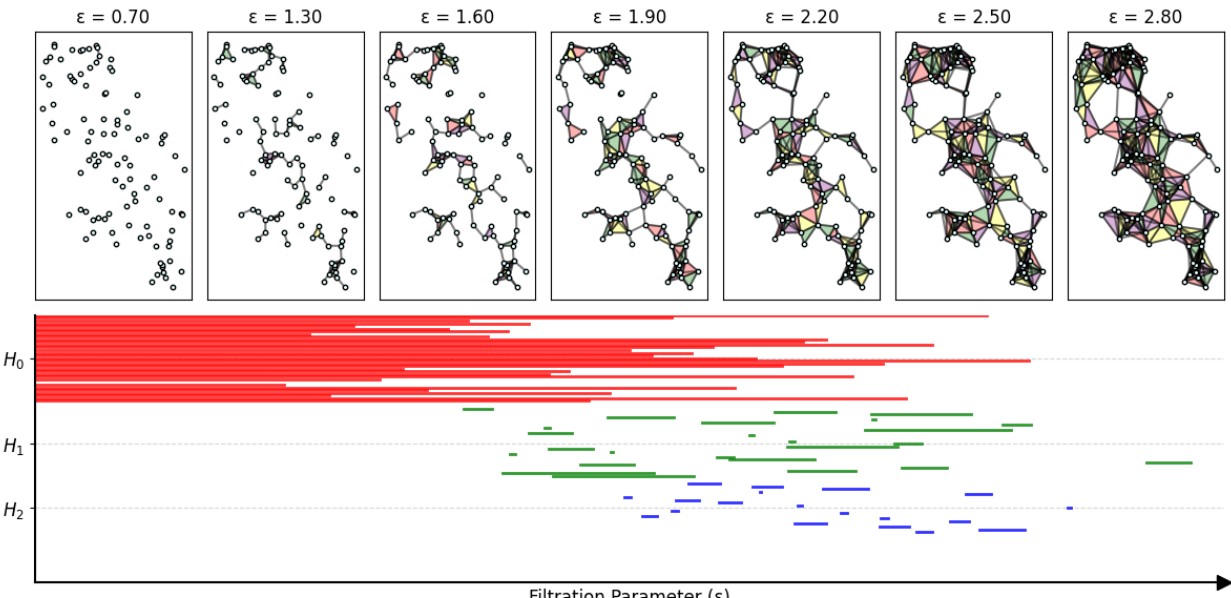

Figure 4: Visualization of topology-aware analysis through Vietoris–Rips filtration. The top row depicts the evolution of the simplicial complex as the filtration parameter $\epsilon$ increases. The bottom part shows the corresponding barcode representation of persistent homology groups in different dimensions ($H_0, H_1, H_2$).

distilled student successfully restores these unbroken, continuous rings and sharpens semantic boundaries. These qualitative examples show reduced fragmentation of large connected regions and cleaner boundaries, which is consistent with our goal of preserving dominant $H_0$ and $H_1$ structures in the distilled representation.

**t-SNE Visualization.** To showcase category boundaries, we have included a t-SNE visualization of the learned point embeddings (using the same feature level used by our distillation constraints) to illustrate how well different categories separate in the feature space. Concretely, we sampled points from the validation set and extracted the pre-classifier embeddings for (i) the teacher, (ii) the student baseline with standard KD, and (iii) our full method. The t-SNE plot shows that the standard-KD student exhibits feature over-

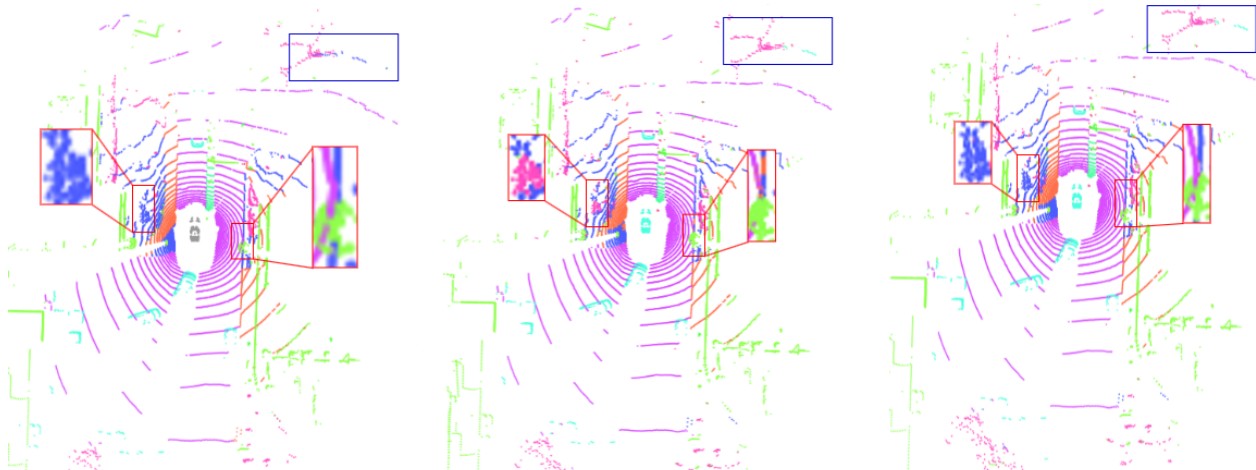

Figure 5: **Qualitative Comparison.** (Left) Ground Truth, (Middle) Student with standard KD, and (Right) Student with our full method (topology and gradient-guided KD). The baseline student produces fragmented predictions on continuous manifolds, whereas our Persistence-Weighted Chamfer loss corrects these structural errors by enforcing topological consistency. Red boxes highlight regions of improved continuity and boundary sharpness, while blue boxes highlight failure cases in extremely sparse regions.

smoothing, resulting in ambiguous, overlapping category boundaries. By introducing our gradient-guided Feature Alignment ($\mathcal{L}_{\mathrm{grad}}$), the student focuses its representational capacity on high-gradient boundary regions, which discourages feature blending and leads to noticeably sharper and more distinct category boundaries. Moreover, the global structure of the student's latent manifold more closely mirrors the teacher, consistent with our topological loss ($\mathcal{L}_{\mathrm{topo}}$) transferring global feature-space geometry without fragmentation.

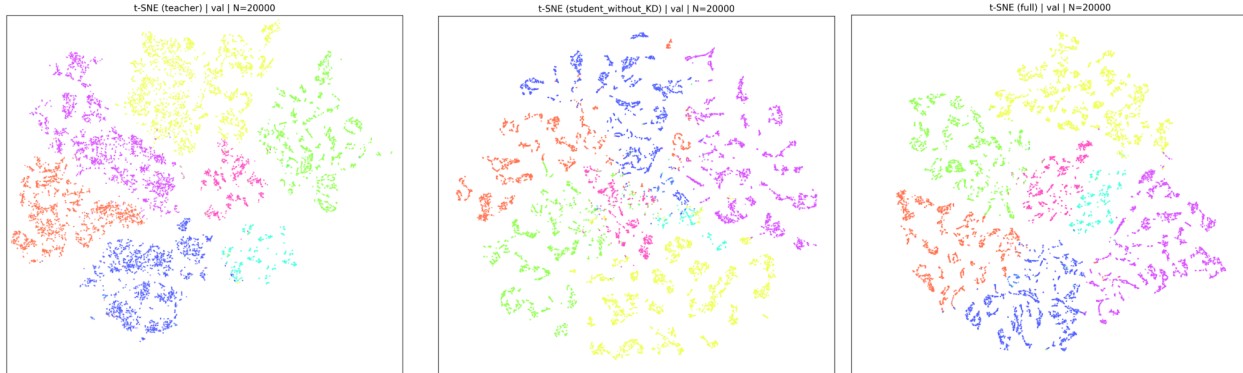

Figure 6: **t-SNE Visualization of Feature Embeddings.** (Left) Teacher, (Middle) Student with standard KD, and (Right) Student with our full method. Our approach yields sharper and more distinct category boundaries compared to the standard KD baseline by discouraging feature blending.

**Failure Case Analysis.** We also highlight and discuss scenarios where our framework currently struggles (indicated by the blue boxes in Figure 5). At extreme ranges, LiDAR points become very sparse, so local neighborhoods become unreliable and the Vietoris–Rips filtration yields few informative 1D structures ($H_1$) beyond trivial disconnected components ($H_0$). In these sparse regions, the topological term therefore provides weaker geometric supervision, and the student can struggle similarly to the baseline. We include a discussion of this sparsity limitation, as well as instances of topological ambiguity (where distinct semantic classes can share similar geometric topologies), which can lead to localized misclassifications.

### D.3 Topological Sensitivity

**Homology Dimension ($H_1$ vs. $H_2$).** We restrict our experiments to $H_0$ and $H_1$. As shown in Figure 8, extending to $H_2$ causes factorial runtime growth due to the combinatorial explosion of 2-simplices (triangles) required for the filtration. Specifically, while the number of edges scales quadratically, the number of triangles scales cubically, causing the boundary matrix reduction step to become a computational bottleneck. Our benchmark reveals that for inputs as small as $N = 400$, computing $H_2$ incurs a latency spike orders of magnitude higher than $H_1$, making it infeasible for iterative training. Furthermore, for surface-based 2.5D LiDAR data, $H_2$ (voids) offers diminishing returns compared to clustering ($H_0$) and loops ($H_1$). Since LiDAR scans capture the *boundary* of objects rather than their volume, genuine enclosed voids are rarely observed; the topological signal is predominantly carried by 1-dimensional cycles (e.g., window frames, poles, and tires), which are fully captured by $H_1$.

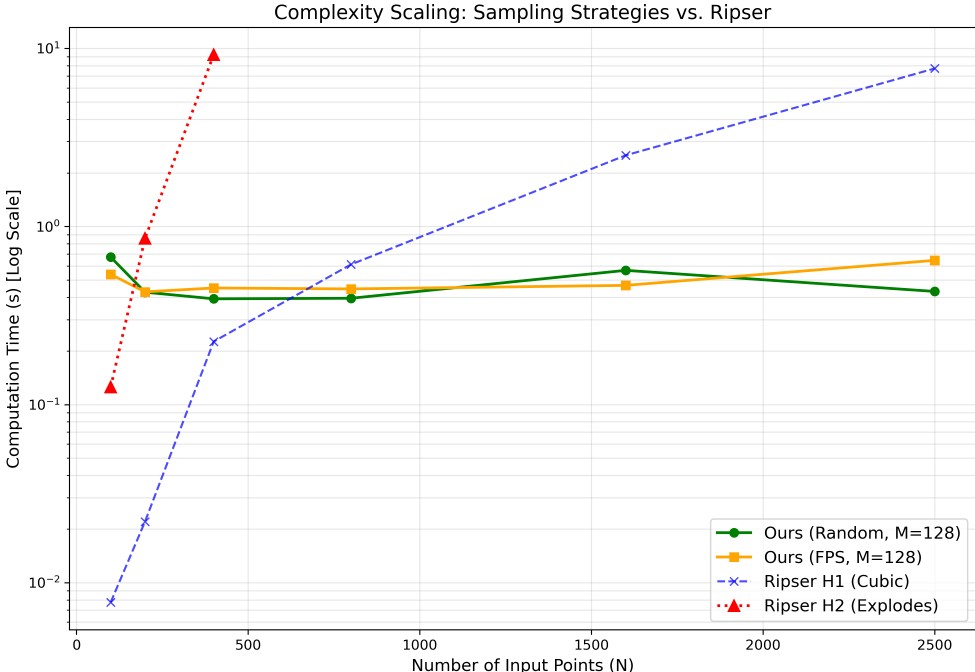

Figure 7: **Complexity Scaling of Subsampling Strategies.** Comparison of computation time between Topology-Preserving Sampling (TPS) as a function of input point cloud size $N$, with a fixed subsample size $M = 128$. Random sampling maintains nearly constant time, whereas TPS scales linearly with $N$, highlighting the trade-off between geometric coverage and computational efficiency for large inputs.

**Noise Mitigation.** Standard topological loss functions are often hypersensitive to sampling noise, treating transient, short-lived features with the same importance as robust structural loops. To prevent the student from overfitting to these artifacts, we introduce a logarithmic weighting function, $w(p) = \log(1 + \text{pers}(p))$, which scales the gradient magnitude based on feature persistence. As demonstrated in Figure 9, this function effectively suppresses the gradients for low-persistence noise ($p \approx 0$) while preserving strong training signals for high-persistence structural features. This ensures the student minimizes the impact of sampling anomalies and focuses solely on learning the robust manifold topology.

### D.4 Gradient Alignment Sensitivity

Our current formulation computes gradient-based saliency with respect to hidden feature activations and aligns teacher/student saliency maps across multiple layers from the decoder. As shown in Table 10, attempting gradient alignment on the encoder stages degrades performance compared to the baseline, as early-stage

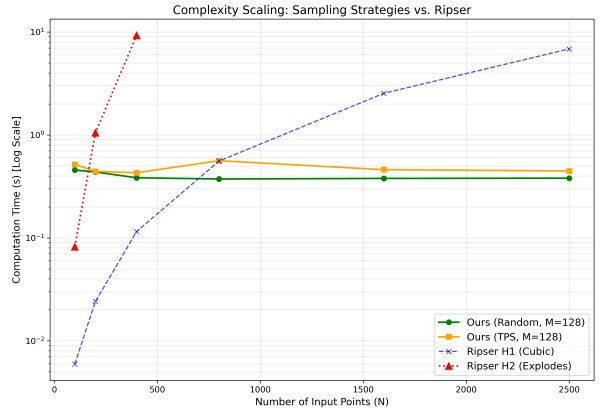
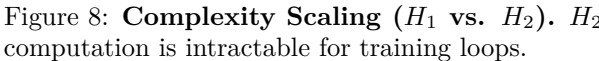

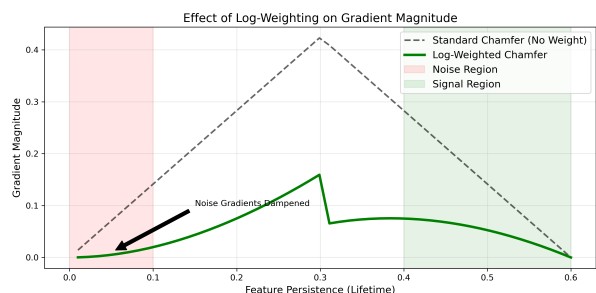

Figure 9: **Gradient Dampening.** Log-weighting suppresses noise (red) vs signal (green).

Figure 8: **Complexity Scaling ($H_1$ vs. $H_2$).** $H_2$ computation is intractable for training loops.

gradients are too diffuse and lack task-specific semantic focus. Conversely, alignment within the decoder stages yields significant gains. While aligning a single early-decoder layer performs strongly on simple geometric abstractions, our full method utilizes multi-scale alignment across all decoder layers. This multi-scale approach proves essential for complex, real-world datasets like nuScenes, as it combines the global semantic context from early decoder layers with the fine-grained boundary refinement of late decoder layers, yielding the best overall robustness and a peak mIoU of 78.2%.

| Gradient Source Layers | mIoU (%) |
| --- | --- |
| None (Topo + KLD only) | 77.6 |
| Encoder Early (enc1-enc2) | 76.5 |
| Encoder Late (enc5) | 76.8 |
| Decoder Early (dec1) | 77.9 |
| Decoder Late (dec4) | 77.7 |
| All Decoder (dec1-dec2-dec3-dec4) | **78.2** |

Table 10: Ablation on Gradient Selection Layers (nuScenes). We compare the impact of extracting gradient-guided saliency maps ($\mathcal{L}_{grad}$) from different subsets of the encoder and decoder stages.

### D.5 Effectiveness of Persistence-Weighted Chamfer Distance over advanced Optimal Transport Algorithms

To justify our use of the Persistence-Weighted Chamfer Distance (PWCD) over advanced Optimal Transport methods, we replaced PWCD with entropically regularized Sinkhorn OT Cuturi (2013) in our topological distillation module ($\mathcal{L}_{\text{topo}}$). Because Sinkhorn is highly sensitive to the entropic regularization parameter $\epsilon$, we swept multiple values.

As shown in Table 11, with tuned $\epsilon$, Sinkhorn OT can slightly improve over the baseline, but it is sensitive to $\epsilon$ and can underperform for other values. We attribute this to gradient diffusion. Diagnostic tracking during training reveals that Sinkhorn's soft-matching creates high gradient entropy when the student's persistence diagram is noisy. Instead of suppressing noise, it partially matches noisy student features to distant teacher features. Conversely, PWCD acts as a hard nearest-neighbor assignment with a diagonal fallback, yielding highly localized gradients (lower entropy) that safely push topological noise to zero persistence. Furthermore, PWCD achieves this increased performance without requiring the exhaustive hyperparameter tuning inherent to Unbalanced OT.

| Topological Distance Metric | Hyperparameters | Gradient Entropy ↓ | mIoU (nuScenes) ↑ |
|---|---|---|---|
| None (CE + KD Baseline) | - | - | 76.9 |
| Sinkhorn OT | $\epsilon = 0.02$ | 0.93 | 76.8 |
| Sinkhorn OT | $\epsilon = 0.05$ | 0.88 | 77.0 |
| Sinkhorn OT | $\epsilon = 0.10$ | 0.82 | 77.2 |
| Sinkhorn OT | $\epsilon = 0.20$ | 0.84 | 77.1 |
| **PWCD (Ours)** | **Parameter-free** | **0.75** | **77.6** |

Table 11: Ablation comparing Sinkhorn Optimal Transport to our Persistence-Weighted Chamfer Distance. Lower gradient entropy indicates more localized, stable gradient routing.

### D.6 Validation of Topological Correctness

To ensure the validity of our custom PyTorch implementation, we verified that it produces persistence diagrams topologically equivalent to the C++ `Ripser` library Bauer (2021). Figure 10 demonstrates this alignment across different geometric primitives.

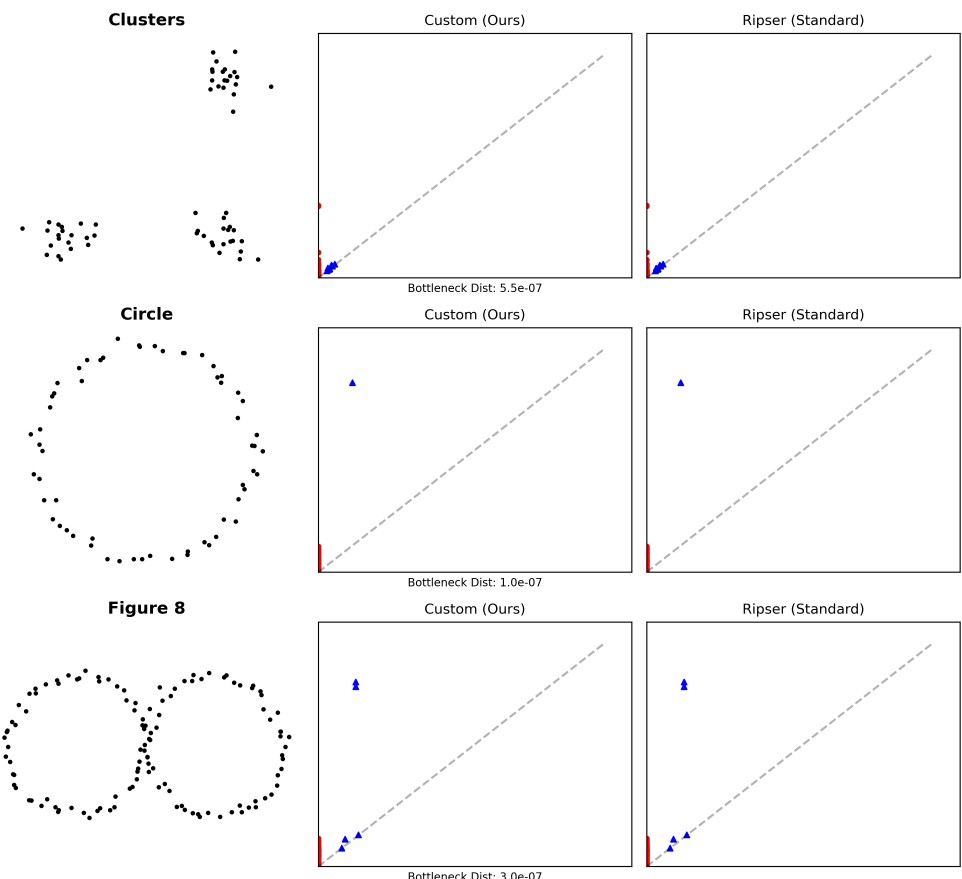

Figure 10: **Validation of Topological Correctness.** Visual comparison of persistence diagrams generated by our pure-PyTorch baseline versus `Ripser`. The diagrams are topologically equivalent with negligible bottleneck distances.

