# OpenReview forum: "Topology- and Gradient-Guided Knowledge Distillation for Point Cloud Semantic Segmentation"
_TMLR — Accepted by TMLR_

### Review · Reviewer_HSVU · 2026-02-10

**Summary Of Contributions:**

Acknowledging the fact that deploying high-performance point cloud models to edge devices is hard, the authors propose a novel distillation framework that leverages topology-aware representations and gradient-guided knowledge distillation to effectively transfer knowledge from a high-capacity teacher to a lightweight student model.

**Audience:**

Yes

**Audience Explanation:**

Yes, I agree that some evidence should be interesting to TMLR's audience, especially in the area that need compressed model for point cloud analysis.

**Claims And Evidence:**

Yes

**Claims Explanation:**

Some designs of the proposed methods remain unclear to me. See "Requested Changes". The claim for the newly proposed TDA supervision lacks sufficient evidence and a clear rationale.

Some discussions on OT are incomplete and confusing.

To me, the paper is imcomplete and some of the critical points should be reframed to include all related discussions.

**Requested Changes:**

1. The motivation of the point cloud-specific knowledge distillation remains unclear to me. As the authors mentioned in Sec. 2, point-based methods can use a standard attention mechanism to process. Then why not use traditional approaches [1-3] that are proven to be sufficient for KD remains unclear to me. Some of the feature-level supervision is very similar to the mentioned approaches.

2. The first question naturally raises my second question on topological data analysis. As claimed in Sec 3.2.1, "The core challenge in integrating Topological Data Analysis (TDA) into deep learning is the discrete nature of the persistence map." In a space that is already continuous (latent space), is it reasonable to proceed with this sense of analysis? Furthermore, is it sufficient or proven to be true that TDA still holds in high-dimensional space?

3. The scale of models should also be reported in the main tables for completeness (e.g., table 1, 2).

4. Teaching-assistant models' discussions are not included. The authors have made a high compression of the teacher model. Under such cases, typically a teaching assistant would be involved to mitigate the large gap between teacher and student models. It is okay the proposed method does not use such designs. However, the discussions should be included.

5. The discussions on the computational complexity of OT are misleading. I understand that the classic OT is (O(n3)). However, more advanced OTs have solved the problem [4-7]. The lack of discussions and the heavy claim as the motivation of Chamfer distance is unreasonable to me.

[1] Ad-kd: Attribution-driven knowledge distillation for language model compression

[2] AMD: Automatic Multi-step Distillation of Large-scale Vision Models

[3] Minilm: Deep self-attention distillation for task-agnostic compression of pre-trained transformers

[4] Sinkhorn distances: Lightspeed computation of optimal transport

[5] Visual recognition with deep nearest centroids

[6] Rethinking semantic segmentation: A prototype view

[7] Optimal transport aggregation for visual place recognition

---

> ### Author Response · Authors · 2026-02-27
> **Response to Reviewer HSVU**
>
> We thank the reviewer for the sharp questions. We have incorporated the requested clarifications and expanded discussions into the revised manuscript. As the discussion phase does not permit uploading a revised PDF or including additional figures in the rebuttal, we summarize the added clarifications below.
>
> > **1. The motivation of the point cloud-specific knowledge distillation remains unclear to me. As the authors mentioned in Sec. 2, point-based methods can use a standard attention mechanism to process. Then why not use traditional approaches [1-3] that are proven to be sufficient for KD remains unclear to me. Some of the feature-level supervision is very similar to the mentioned approaches**
>
> We agree that modern point-based networks (including our PTv3 teacher) use standard self-attention, so in principle attention/feature distillation methods such as AD-KD [1], AMD [2], and MiniLM [3] are relevant baselines.
> Our clarification is that the motivation is not that point cloud models cannot use traditional KD. Rather, standard KD objectives alone do not explicitly preserve the global geometric continuity and critical regions that are commonly degraded in heavily compressed 3D segmentation students.
> * **We already include a standard, traditional KD baseline.** Our baseline uses logit distillation (KL) + segmentation loss, following the standard KD method (task loss and KL-divergence only). On three datasets, this baseline reaches 76.9 / 71.8 / 67.9 mIoU on nuScenes / SemanticKITTI / Waymo (Table 7 in the submission).
> * **Why we add topology and gradient guidance.** Under strong compression, student errors in 3D semantic segmentation often manifest as (i) fragmentation of large continuous regions (loss of connectedness) and (ii) boundary over-smoothing for thin or small structures. These are precisely the kinds of issues that topology-aware losses have been shown to matter in dense prediction settings (e.g., topology-preserving segmentation losses [13], which inspired our design).
> * **Empirical evidence that traditional KD is helpful but insufficient.** Table 7 shows that adding topology and gradient-guided alignment consistently improve over standard KD alone.
>
> In the revised version, we have incorporated the following clarifications: **(i)** we explicitly position our method as complementary to standard KD/attention KD methods [1]–[3], **(ii)** we clarify that our primary contribution is transferring structural topology and gradient-guided alignment, and **(iii)** we add a short discussion contrasting our approach with existing topological KD methods such as TGD [14] and TopKD [15], highlighting that our setting targets large-scale LiDAR segmentation and leverages diagram-level matching with custom gradient routing rather than persistent-image vectorizations.
>
> ---
>
> > **2. The first question naturally raises my second question on topological data analysis. As claimed in Sec 3.2.1, "The core challenge in integrating Topological Data Analysis (TDA) into deep learning is the discrete nature of the persistence map." In a space that is already continuous (latent space), is it reasonable to proceed with this sense of analysis? Furthermore, is it sufficient or proven to be true that TDA still holds in high-dimensional space?**
>
> We appreciate the reviewer’s comment and agree that our current phrasing can be clearer.
>
> **(a) Is TDA reasonable in a continuous latent space?**
>
> Yes. Although the ambient latent space $\mathbb{R}^d$ is continuous, for each input the network produces a finite set of feature vectors (a point set in $\mathbb{R}^d$). Persistent homology is explicitly designed to infer topological structure from such finite samples in a metric space. This viewpoint (computing PH on latent embeddings) is widely used in topology-aware deep learning (e.g., Topological Autoencoders [9], Perslay [10], topology layers [11])
>
> **(b) What did we mean by “discrete nature of the persistence map”?**
>
> Our intended meaning is optimization/differentiation, not that latent spaces being continuous makes TDA invalid. Persistent homology computation involves discrete algorithmic steps (simplex ordering + matrix reduction). However, for Vietoris–Rips filtrations, birth/death times are piecewise-smooth functions of the pairwise distances as long as the persistence pairing and maximizers are locally stable; non-smoothness occurs at degeneracies (pairing changes or max-edge ties). This is the basis used by prior differentiable PH approaches ([9]-[12]). Our implementation uses Ripser [12] in the forward pass and routes gradients through the critical edges of the paired simplices, following this established principle.

---

> > ### Author Response · Authors · 2026-02-27
> > **Response to Reviewer HSVU (continue 1)**
> >
> > **(c) Does TDA “hold” in high-dimensional space?**
> >
> > Persistent homology is defined for any metric space and depends only on the pairwise distance matrix, so it is mathematically well-defined independent of the ambient dimension. The more subtle question is whether it remains informative when distances become noisy in high dimensions. We address this in our framework by: **(1)** applying PH to learned embeddings whose geometry is shaped by the task; **(2)** using Topology-Preserving Sampling with a stability/interleaving argument (Appendix A.2, building on PH stability ideas [16]), and **(3)** suppressing near-diagonal artifacts through persistence weighting (Appendix A.3), which reduces the impact of low-persistence noise.
> >
> > In the revised manuscript, we have rewritten Section 3.2.1 to make these three points explicit and remove any wording that could be misconstrued.
> >
> > ---
> >
> > > **3. Teaching-assistant models' discussions are not included. The authors have made a high compression of the teacher model. Under such cases, typically a teaching assistant would be involved to mitigate the large gap between teacher and student models. It is okay the proposed method does not use such designs. However, the discussions should be included**
> > We thank the reviewer for this important point. We agree that multi-step distillation with an intermediate teacher assistant is a standard solution for large capacity gaps (e.g., TAKD [8] and also discussed in the distillation literature, including [2] [3]).
> >
> > Our rationale for not using a TA in this submission is practical and methodological:
> > * **Same-family backbone and strong student baseline.** Our student is a scaled PTv3 variant (same design family), and it achieves a competitive non-distilled baseline rather than suffering training collapse. In this regime, the main limitation is not optimization stability but what information is transferred under compression.
> > * **We transfer structural signals that reduce the need for an intermediate model.** Instead of adding extra networks, we add constraints that directly target **(i)** global manifold/topological structure $L_{topo}$ and **(ii)** gradient-guided alignment $L_{grad}$, which empirically improve results across datasets on top of standard KD (Table 7).
> >
> > That said, we fully agree a discussion is warranted. In the revised manuscript, we have added a dedicated paragraph explaining when TA distillation is typically beneficial, and we note that TA could be used in future work to further smooth the transfer in even more extreme compression settings.
> >
> > ---
> >
> > > **4. The discussions on the computational complexity of OT are misleading. I understand that the classic OT is (O(n3)). However, more advanced OTs have solved the problem [4-7]. The lack of discussions and the heavy claim as the motivation of Chamfer distance is unreasonable to me**
> >
> > The reviewer is right that our current wording around OT complexity is oversimplified: while exact OT/assignment formulations can have cubic complexity, entropic regularization (Sinkhorn) and other approximations significantly improve practical runtime and are widely used [7].
> >
> > In the revised manuscript, we have **(i)** explicitly acknowledged Sinkhorn-style OT and the reviewer’s references [4]–[7], and **(ii)** reframed our motivation: our choice of a Chamfer-style objective is driven primarily by robustness and gradient behavior for persistence diagrams, not solely by asymptotic runtime.
> >
> > Our technical rationale is:
> > * **Dynamic, unequal cardinalities and heavy near-diagonal noise in PDs.** In distillation, the student PD can contain many low-persistence points early in training, and the teacher/student PD sizes can differ substantially and vary across samples/batches. Any transport-based metric must decide how to handle unmatched mass (often via diagonal projection, unbalanced OT, or partial OT). This introduces additional modeling choices and hyperparameters.
> > * **Gradient stability is the key issue in our setting.** Even if Sinkhorn makes OT fast, it can produce a global soft matching. When the student PD is noisy, this can distribute gradients broadly across many low-value matches, making early optimization sensitive to regularization and mass-relaxation parameters. In contrast, our nearest-neighbor Chamfer formulation with diagonal fallback yields localized, direct gradients that **(a)** naturally tolerate unequal set sizes and **(b)** allow unmatched/noisy student points to be pushed toward the diagonal (vanishing persistence) without forcing global mass conservation.
> > * **We further suppress teacher noise via persistence weighting.** Our log persistence weighting provably reduces the influence of near-diagonal teacher points (Appendix A.3, Lemma 1), ensuring the student is not incentivized to mimic stochastic low-persistence artifacts.

---

> ### Author Response · Authors · 2026-02-27
> **Response to Reviewer HSVU (continue 2)**
>
> **(Question 4. cont)**
>
> Regarding Unbalanced/Partial OT (UOT/POT): we agree these are principled tools for unequal cardinalities. In the revised manuscript, we have added a short discussion (with citations such as [17] and [18]) noting that UOT/POT introduce additional hyperparameters (e.g., entropic regularization and mass-divergence penalties) whose tuning can be nontrivial when PD size/noise fluctuates across training. Our Persistence-Weighted Chamfer Distance was chosen as a stable, parameter-light alternative for this specific KD setting.
>
> ---
>
> > **5. Report model scale in main tables**
>
> In the revised manuscript, we have added Params and FLOPs (and/or equivalent compute measures) directly to the main comparison tables (e.g., Tables 1–2), in addition to the existing params/FPS reporting in Table 3. For prior works, we report model scale where the original papers or official implementations provide sufficient detail; otherwise, we mark values as not reported rather than speculate.
>
> ---
>
> **References:**
>
> [1] Wu et al. AD-KD: Attribution-Driven Knowledge Distillation for Language Model Compression. ACL 2023.
>
> [2] Han et al. AMD: Automatic Multi-step Distillation of Large-Scale Vision Models. ECCV 2024.
>
> [3] Wang et al. MiniLM: Deep Self-Attention Distillation for Task-Agnostic Compression of Pre-Trained Transformers. NeurIPS 2020.
>
> [4] Cuturi. Sinkhorn Distances: Lightspeed Computation of Optimal Transport. NeurIPS 2013.
>
> [5] Wang et al. Visual Recognition with Deep Nearest Centroids. ICLR 2023.
>
> [6] Zhou et al. Rethinking Semantic Segmentation: A Prototype View. CVPR 2022.
>
> [7] Izquierdo & Civera. Optimal Transport Aggregation for Visual Place Recognition. CVPR 2024.
>
> [8] Mirzadeh et al. Improved Knowledge Distillation via Teacher Assistant. AAAI 2020.
>
> [9] Moor et al. Topological Autoencoders. ICML 2020.
>
> [10] Carrière et al. PersLay: A Neural Network Layer for Persistence Diagrams. AISTATS 2020.
>
> [11] Brüel-Gabrielsson et al. A Topology Layer for Machine Learning. AISTATS 2020.
>
> [12] Bauer. Ripser: Efficient Computation of Vietoris–Rips Persistence Barcodes. J. Appl. Comput. Topol. 2021.
>
> [13] Hu et al. Topology-Preserving Deep Image Segmentation. NeurIPS 2019.
>
> [14] Jeon et al. Leveraging Topological Guidance for Improved Knowledge Distillation (TGD). 2024.
>
> [15] Kim et al. Do Topological Characteristics Help in Knowledge Distillation? (TopKD). ICML 2024.
>
> [16] Cohen-Steiner et al. Stability of Persistence Diagrams. Discrete & Computational Geometry 2007.
>
> [17] Chizat et al. Scaling Algorithms for Unbalanced Optimal Transport Problems. Mathematics of Computation 2018
>
> [18] Fatras et al. Unbalanced Minibatch Optimal Transport; Applications to Domain Adaptation. ICML 2021.

---

> > ### Comment · Reviewer_HSVU · 2026-02-27
> >
> > I appreciate the authors' detailed response, which has mostly solved my concerns.
> >
> > One additional comment will be a comparison between OT and the Chamfer distance, which can further strengthen the claim in Q4.

---

> > > ### Author Response · Authors · 2026-03-17
> > > **Response to Reviewer HSVU**
> > >
> > > Dear Reviewer HSVU,
> > >
> > > Thank you for your constructive questions. We have added a table comparing Optimal Transport (OT) and Chamfer distance (Table 11) in the latest revision.
> > >
> > > Best regards,
> > >
> > > The Authors

---

### Review · Reviewer_Wmca · 2026-02-22

**Summary Of Contributions:**

The authors introduce a novel distillation framework that combines topology-aware knowledge representation with gradient-guided distillation, addressing the challenges of deploying high-performance point cloud models in resource-constrained environments.
The proposed framework exploits the intrinsic geometric and structural characteristics of point clouds by incorporating topological information into the distillation process, thereby preserving critical features essential for accurate predictions.
By integrating gradient-guided distillation, the approach selectively highlights salient features, facilitating efficient and effective knowledge transfer from the teacher model to the student model.
The proposed method is verified to be effective across multiple datasets.

**Audience:**

Yes

**Audience Explanation:**

Yes. The paper addresses knowledge distillation and efficient model design, which are topics of broad interest to the TMLR community. While the empirical focus is on point cloud models, the proposed topology-aware and gradient-guided distillation framework introduces methodological insights that may extend to other structured data modalities and resource-constrained learning settings.

**Broader Impact Concerns:**

No concerns.

**Claims And Evidence:**

Yes

**Claims Explanation:**

The reviewer has checked the claims and they looks ok.

**Requested Changes:**

1. Lack of discussion regarding other point cloud feature extraction frameworks, e.g., [1], [2].

[1]  "MASS: Multi-attentional semantic segmentation of LiDAR data for dense top-view understanding." IEEE Transactions on Intelligent Transportation Systems 23.9 (2022): 15824-15840.

[2] "Pillarsegnet: Pillar-based semantic grid map estimation using sparse lidar data." arXiv preprint arXiv:2105.04169 (2021).

2. Lack of qualitative results and failure cases. The authors are suggested to visualize the predictions of some baselines and the proposed approach to deliver more qualitative analysis.


3. TSNE visualization is recommended to be added to showcase the category boundaries.


4. In Eq. 8, the authors leveraged different weights for different loss function. How did the authors select the hyperparameters and how sensible those hyperparameters are across dataset?


5. In Section 4.1, more insights regarding the methodology benefits should be enriched. The authors only briefly mentioned that the proposed method achieves better performance which lacks of in-depth discussion.

6. In Section 3.1.1, the gradients of the task loss are computed with respect to the hidden feature activations. An ablation study examining the impact of selecting gradients from different layers would further clarify the contribution of this design choice.

---

> ### Author Response · Authors · 2026-02-27
> **Response to Reviewer Wmca**
>
> Dear Reviewer Wmca,
>
> We have incorporated the requested citations, qualitative analyses, and clarifications into the revised manuscript. As the discussion phase does not permit uploading a revised PDF or including additional figures in the rebuttal, we summarize the added analyses below; the corresponding figures and detailed discussions are included in the revised version.
>
> > **1. Lack of discussion regarding other point cloud feature extraction frameworks**
>
> We thank the reviewer for pointing out MASS and PillarSegNet, which are representative BEV/pillar/top-view feature extraction pipelines for LiDAR understanding. We have properly cited and discussed the suggested works in our revised version.
>
> Although there are some similarities, our work is different from these feature extraction/backbone choices: we focus on knowledge distillation (KD) to transfer both **(i)** global geometric/topological structure and **(ii)** task-relevant spatial saliency from a strong teacher to a lightweight student. In addition, while our topological loss is formulated in feature space and is architecturally agnostic in principle (as long as we can obtain a set of feature vectors and a meaningful metric), applying it across fundamentally different spatial representations (point &harr; BEV/pillar) requires nontrivial alignment (scatter/gather) that would confound an isolated evaluation of the loss. Furthermore, PTv3 operates on raw point tokens, whereas MASS utilizes a dense top-view grid and PillarSegNet uses a sparse pillar grid. Distilling knowledge from a high-capacity point-based teacher (like PTv3) to a grid- or pillar-based student would require additional spatial feature alignment modules (e.g., point-to-voxel or point-to-pillar scattering/gathering) before the topological loss can be computed.
>
> To rigorously validate our core contribution, we specifically chose a homogeneous point-to-point distillation setup. This isolated our proposed losses from the confounding variables introduced by cross-representation feature mapping. However, we completely agree that applying our feature-space topological distillation to intra-architecture setups (e.g., PillarSegNet-Teacher to PillarSegNet-Student) or cross-representation setups is a highly valuable next step. We have explicitly noted the extension of this paradigm to multi-view and alternative sensor fusion systems as a primary direction in our Future Work section.
>
> ----
>
> > **2. Lack of qualitative results and failure cases. The authors are suggested to visualize the predictions of some baselines and the proposed approach to deliver more qualitative analysis.**
>
> We agree that additional qualitative evidence and explicit failure cases strengthen the paper. In the revised manuscript, we have added qualitative visualizations and an explicit failure-case discussion. Specifically, we have highlighted cropped regions showing how the baseline student (with standard KD) produces fragmented predictions on continuous manifolds (e.g., terrain or large vehicles), and how our Persistence-Weighted Chamfer loss corrects these structural errors by enforcing topological consistency.
>
> In the revised manuscript, we have included a qualitative comparison (left-to-right: Ground Truth, Student with standard KD, and Student with our full method: topology + gradient-guided KD). We summarize the key observations below:
>
> *  **Additional Qualitative Results:** We visualize cropped regions where the baseline lightweight student severely fractures continuous spatial structures and exhibits boundary discontinuation. Our distilled student successfully restores these unbroken, continuous rings and sharpens semantic boundaries. These qualitative examples show reduced fragmentation of large connected regions and cleaner boundaries, which is consistent with our goal of preserving dominant H0/H1 structures in the distilled representation.
> * **Failure Case Analysis (Blue Box):** We also highlight and discuss scenarios where our framework currently struggles. At extreme ranges, LiDAR points become very sparse, so local neighborhoods become unreliable and the Vietoris–Rips filtration yields few informative 1D structures (H1) beyond trivial disconnected components (H0). In these sparse regions, the topological term therefore provides weaker geometric supervision, and the student can struggle similarly to the baseline. We include a discussion of this sparsity limitation, as well as instances of topological ambiguity (where distinct semantic classes can share similar geometric topologies), in the revised manuscript.

---

> > ### Author Response · Authors · 2026-02-27
> > **Response to Reviewer Wmca (continue 1)**
> >
> > > **3. TSNE visualization is recommended to be added to showcase the category boundaries**
> >
> > Thank you for the suggestion. In our revised paper, we have included a t-SNE visualization of the learned point embeddings (using the same feature level used by our distillation constraints) to illustrate how well different categories separate in the feature space.
> >
> > Concretely, we sampled points from the validation set and extracted the pre-classifier embeddings for **(i)** the teacher, **(ii)** the student baseline with standard KD, and **(iii)** our full method. In the revised manuscript, the t-SNE plot shows that the standard-KD student exhibits feature over-smoothing, resulting in ambiguous, overlapping category boundaries. By introducing our gradient-guided Feature Alignment ($L_{grad}$), the student focuses its representational capacity on high-gradient boundary regions, which discourages feature blending and leads to noticeably sharper and more distinct category boundaries. Moreover, the global structure of the student’s latent manifold more closely mirrors the teacher, consistent with our topological loss ($L_{topo}$) transferring global feature-space geometry without fragmentation.
> >
> > ---
> >
> > > **4. In Eq. 8, the authors leveraged different weights for different loss function. How did the authors select the hyperparameters and how sensible those hyperparameters are across dataset?**
> >
> > We sincerely thank the reviewer for raising this important point. We agree that providing clarity on the selection process and sensitivity of the loss weights is essential for evaluating the robustness of our distillation framework.
> >
> > Exhaustively grid-searching the optimal distillation parameters across all three large-scale datasets from scratch is highly computationally demanding. Therefore, our selection strategy was to perform a targeted sweep using the nuScenes training split, selecting weights by validation mIoU; then we reused the same weights unchanged for SemanticKITTI and Waymo. We evaluated a grid where $\lambda_{topo} \in \{0.5, 4.0, 8.0\}$, and $\lambda_{grad}, \lambda_{kd} \in \{0.5, 1.0, 2.0\}$. The optimal configuration discovered on nuScenes ($\lambda_{topo}=8.0$, $\lambda_{grad}=1.0$, $\lambda_{kd}=1.0$) was subsequently applied to SemanticKITTI and Waymo.
> >
> > We thank the reviewer for this prompt; adding this transparency significantly strengthens the empirical foundation of the paper. We have included these details in the revised manuscript.
> >
> > ---
> >
> > > **5. In Section 4.1, more insights regarding the methodology benefits should be enriched. The authors only briefly mentioned that the proposed method achieves better performance which lacks of in-depth discussion**
> >
> > Thank you for pointing this out. We agree that the original Sec. 4.1 was too brief and mainly performance-oriented. In the revised manuscript, we have expanded Sec. 4.1 to provide mechanism-level insight into what each component enforces and how that maps to observable behavior in point cloud semantic segmentation, supported directly by the newly added qualitative and t-SNE visualizations.
> >
> > Under strong compression, a lightweight student often exhibits two systematic failure modes in 3D segmentation: **(1)** structural fragmentation (large continuous regions break into scattered islands/holes), and **(2)** class boundaries are over-smoothed. These issues are not fully addressed by matching logits or local feature statistics alone, since they are global-structure and boundary-saliency problems.
> >
> > * **Topology-aware distillation improves global structural coherence (less fragmentation).** Our topological supervision $L_{topo}$ distills the teacher’s persistent topological structures computed from feature embeddings, focusing on: (a) H0 (connected components): encourages the student to preserve the teacher’s dominant connectedness patterns. This mitigates the feature over-smoothing that causes localized fragmentation on continuous spatial manifolds; (b) H1 (cycles/loops): encourages the preservation of non-trivial structural patterns that are often lost or washed out under aggressive compression. As shown by the qualitative visualizations added to the revised manuscript, the baseline lightweight student (middle) severely fractures continuous spatial structures (e.g., scattered predictions on large manifolds), while our topology-guided student (right) restores these into more continuous semantic regions.
> > * **Persistence weighting prevents the student from chasing topological noise.** Persistence diagrams contain many near-diagonal (low-persistence) points that typically correspond to unstable/local noise. Treating all points equally can encourage the student to imitate artifacts and waste capacity. Our persistence weighting isolates the critical simplices, ensuring the distillation focuses strictly on dominant, stable structures.

---

> > > ### Author Response · Authors · 2026-02-27
> > > **Response to Reviewer Wmca (continue 2)**
> > >
> > > * **Gradient-guided distillation improves boundary-sensitive and hard regions (sharper boundaries).** Our $L_{grad}$ aligns teacher/student task-loss gradient saliency with respect to hidden activations, which highlights regions where the task loss is most sensitive, typically class boundaries, thin structures, and rare/small objects. Aligning these saliency maps encourages the student to allocate its limited capacity to these harder regions that are often over-smoothed under aggressive compression.
> > >
> > > ---
> > >
> > > > **6. Ablation: impact of selecting gradients from different layers**
> > >
> > > Our current formulation computes gradient-based saliency with respect to hidden feature activations and aligns teacher/student saliency maps across multiple layers from later layers in the decoder, which is intended to capture multi-scale discriminative regions rather than relying on a single representation level. We hypothesize that later layers (closer to the segmentation head) produce more semantically meaningful saliency, and combining multiple scales yields the best overall boundary quality and robustness. In our revised paper, we have included an ablation that computes the saliency alignment loss using gradients from different subsets of layers (early/mid/late encoder stages and/or decoder stages).

---

### Review · Reviewer_VoWk · 2026-03-03

**Summary Of Contributions:**

This paper proposes a method for knowledge distillation of point cloud semantic segmentation. The distillation framework uses topology-aware representations and gradient-guided knowledge distillation to distill knowledge from a bigger teacher model to a lightweight student model. The proposed approach claims to "capture the underlying geometric structures of point clouds while selectively guiding the student model's learning process through gradient-based feature alignment."  Experimental results in the Nuscenes, SemanticKITTI, and Waymo datasets demonstrate that the proposed method achieves competitive performance, with an approximately 16x reduction in model size and up to 1.9x decrease in inference time compared to its teacher model.

The proposed method seems effective in resolving knowledge distillation of point cloud semantic segmentation, given the experimental results. However, I feel like the novelty, the addressed tasks, and the scope of this paper are quite limited. The proposed method only targets point cloud segmentation, while nowadays, I believe foundational point cloud models are more promising for all kinds of downstream tasks.

**Audience:**

Yes

**Audience Explanation:**

Maybe? But as I mentioned above, the novelty, the addressed tasks, and the scope of this paper are quite limited. The proposed method only targets point cloud segmentation, while nowadays, I believe foundational point cloud models are more promising for all kinds of downstream tasks.

**Broader Impact Concerns:**

No border impact concerns.

**Claims And Evidence:**

Yes

**Claims Explanation:**

I think the claims made in this paper are reasonable and are well supported.

**Requested Changes:**

I feel like the authors could explore whether the proposed methods can distill foundational point cloud models. If so, the impact of this work will be quite significant.

---

> ### Author Response · Authors · 2026-03-06
> **Response to Reviewer VoWk**
>
> We appreciate the insightful suggestion regarding the distillation of foundational point cloud models. We agree that foundation models represent a highly promising direction for the field. In response, we would like to clarify the foundational nature of our chosen teacher, the generality of our proposed method, and the specific practical scope of our work.
>
> 1. **Point Transformer v3 as a Foundation Architecture.** While our evaluation focuses on a specific downstream task, the teacher model employed in our framework, Point Transformer V3 (PTv3) [1], is widely recognized as a state-of-the-art foundation architecture for 3D point cloud understanding. Due to its massive receptive field, scalability, and representation capacity, PTv3 frequently serves as the core backbone for universal 3D foundation models [2]. By utilizing PTv3, we have effectively demonstrated that our framework can successfully distill knowledge from a modern, high-capacity foundation architecture into a highly efficient, lightweight student model.
>
> 2. **Generality of the Distillation Framework.** We would like to emphasize that our proposed distillation method is neither architecture-specific nor strictly limited to semantic segmentation. The core components of our approach operate fundamentally at the representation level. Because these mechanisms target the underlying geometric structures inherent to all point cloud data, they can be naturally adapted to distill other foundation point cloud models or applied to various dense prediction tasks (e.g., 3D object detection, panoptic segmentation).
>
> 3. **Scope: Deriving Compact Specialists for Deployment.** This paper's goal is not to propose a new general-purpose foundation model, but rather to derive compact models for autonomous driving point cloud segmentation. This problem is practically important and widely studied in real-world benchmarks. Semantic segmentation of point clouds depends heavily on fine-grained geometric structures, and our knowledge distillation method is specifically designed to capture and preserve such geometric patterns, which is a novelty of our work. That said, we believe our approach can be extended to many other point cloud–based tasks, and we leave this direction for future work.
>
> **References:**
>
> [1] Wu, X., Jiang, L., Wang, P. S., Liu, Z., Liu, X., Qiao, Y., Ouyang, W., He, T., & Zhao, H. (2024). "Point Transformer V3: Simpler, Faster, Stronger." *In Proceedings of the IEEE/CVF Conference on Computer Vision and Pattern Recognition (CVPR)*.
>
> [2] Zhu, H., Yang, H., Wu, X., et al. (2025). "PonderV2: Improved 3D Representation with A Universal Pre-training Paradigm." *IEEE Transactions on Pattern Analysis and Machine Intelligence (T-PAMI)*.

---

### Author Response · Authors · 2026-04-14
**Inquiry on Manuscript Status**

Dear Editors,

I hope you are doing well.

I am writing to kindly inquire about the current status of our manuscript.

We have carefully addressed all reviewers' comments and submitted a revised version along with detailed responses. Since it has been weeks since our last update, I would greatly appreciate any information you could share regarding the progress of the evaluation, as well as any further feedback from the reviewers or action editors.

We fully understand that the review process can take time, and we sincerely appreciate the effort and coordination involved. I just wanted to check in to ensure that no additional information is required from our side.

Thank you very much for your time and consideration.

Best regards,
Authors

---

### Decision · Action_Editor_3rCJ · 2026-04-28

**Recommendation:** Accept as is

**Audience:**

Yes

**Audience Explanation:**

Accurate and efficient pointcloud segmentation in outdoor scenes, particularly for autonomous vehicles, is an active area of research and highly practical. One reviewer maintains that "the novelty, the addressed tasks, and the scope of this paper are quite limited", but novelty is not a concern for TMLR, the addressed task is highly relevant to outdoor embodied vision/robotics applications, and narrow-scope papers are often useful.

**Claims And Evidence:**

Yes

**Claims Explanation:**

All three reviewers agree that the claims made in the submission are supported by accurate, convincing and clear evidence. There were initially some concerns about evidence for the topological supervision but this was resolved during rebuttal.